# Decoding individual identity from brain activity elicited in imagining common experiences

Andrew James Anderson 📵 [1,2 ✉], Kelsey McDermott[3,4], Brian Rooks[3], Kathi L. Heffner[3,5,6], David Dodell-Feder 📵 [1,2,7] & Feng V. Lin[1,2,3,8,9]

Everyone experiences common events differently. This leads to personal memories that presumably provide neural signatures of individual identity when events are reimagined. We present initial evidence that these signatures can be read from brain activity. To do this, we progress beyond previous work that has deployed generic group-level computational semantic models to distinguish between neural representations of different events, but not revealed interpersonal differences in event representations. We scanned 26 participants' brain activity using functional Magnetic Resonance Imaging as they vividly imagined them-selves personally experiencing 20 common scenarios (e.g., dancing, shopping, wedding). Rather than adopting a one-size-fits-all approach to generically model scenarios, we con-structed personal models from participants' verbal descriptions and self-ratings of sensory/ motor/cognitive/spatiotemporal and emotional characteristics of the imagined experiences. We demonstrate that participants' neural representations are better predicted by their own models than other peoples'. This showcases how neuroimaging and personalized models can quantify individual-differences in imagined experiences.

[1] Department of Neuroscience, University of Rochester Medical Center, Rochester, NY 14642, USA. [2] Del Monte Institute for Neuroscience, University of Rochester Medical Center, Rochester, NY 14642, USA. [3] Elaine C. Hubbard Center for Nursing Research on Aging, School of Nursing, University of Rochester Medical Center, Rochester, NY 14642, USA. [4] Neuroscience, University of Arizona, Tucson, AZ 85721, USA. [5] Department of Psychiatry, University of Rochester Medical Center, Rochester, NY 14642, USA. [6] Division of Geriatrics and Aging, Department of Medicine, University of Rochester Medical Center, Rochester, NY 14642, USA. [7] Department of Psychology, University of Rochester, Rochester, NY 14642, USA. [8] Department of Neurology, University of Rochester Medical Center, Rochester, NY 14642, USA. [9] Department of Brain and Cognitive Sciences, University of Rochester, Rochester, NY 14642, USA. ✉email: aander41@ur.rochester.edu

Almost everyone can imagine themselves at a wedding, however each person does so differently because they have been to different weddings and hence draw upon memories that no one else has. Our personal history of episodic memories contributes to defining us as individuals and in extreme cases—where memories are of traumatic events—can profoundly affect our psychological health and quality of life. A principal goal of cognitive science is to understand how such memories are represented and manipulated in the human brain. Memories of past experiences can be activated and relived through recollection and are thought to be pieced together to support the mental simulation of hypothetical scenarios[1–5]. Functional brain imaging now enables the systematic study of brain activation elicited during recall and imagination. A long-term vision for the future might therefore be to devise technology that provides a comprehensive neural readout of the information that one voluntarily activates in mental imagery. More humble prerequisites for this are to establish that neural activity elicited during imagery captures meaningful differences between different individuals' mental simulations of similar kinds of events, and to devise quantitative methods that can predict the information represented in neural activation.

In previous work, functional Magnetic Resonance Imaging (fMRI) studies of brain activity have identified a core distributed network of neuroanatomical regions that are reliably activated during the recollection and/or imagination of different experiences and scenarios[6–8]. Regions that are activated in episodic recollection and simulation strongly overlap, which suggests that similar neural machinery is engaged in both cases[1,3–5] (although activation patterns elicited in remembering, and imaging possible past and future events are still distinguishable[5]). This core episodic recollection and simulation network includes regions of medial parietal cortex, inferior parietal cortex, medial prefrontal cortex, and medial and lateral temporal lobe[1–13]. Researchers seeking to decipher what information is represented in brain activity within regions of this network have shown that different types of event can be distinguished from multiple network regions[14–17], as well different components of individual events[18–24] (e.g., people, places, objects, space/time of occurrence). However, it has remained unclear whether differences in activation patterns between individuals imagining similar types of event represent anything more than functionally-irrelevant between-subject noise.

Reason to hypothesize that neural correlates of person-specific imagery may be readable from fMRI data comes in part from a recent thrust of research that has exposed individual differences in other cognitive domains[25]. Prominent examples include: using resting-state fMRI data to predict individual differences in the cortical distribution of activity elicited in multiple (gambling/motor/language etc.) task-related studies[26], matching representational similarity of fMRI elicited viewing photos of personal belongings to behavioral measures of object similarity[27]; and using measures of brain activity scanned during a picture-based mechanical engineering task to predict individuals' physics/engineering exam results[28]. Research has even begun to characterize brain network activity in terms of the nature of individual differences in ongoing thought, reflecting whether current thoughts are detailed, correspond to the past or future, are verbal or in images and so on[29,30]. However, whilst these studies do demonstrate that individual differences can be discerned using fMRI, it is unclear that the methods utilized to elicit individual-differences (e.g., picture interpretation, resting) and detect them (e.g., exam results, object similarity judgments, ratings on components of thought) would generalize to the current case of imagining oneself in multiple different scenarios when cued by generic prompts such as "a wedding" or "a funeral".

In addition, a number of studies have cataloged between-group differences in autobiographical memory-related fMRI activation. For instance, altered activation has been observed in Alzheimer's disease[31,32], semantic dementia[33], and epilepsy[34]. Relating more to conceptual knowledge, machine learning methods have discriminated between young suicidal ideators thinking about suicide and emotion concepts and healthy controls[35], and students contemplating objects' mechanical function at different phases of education[36]. However, whilst all of the previous between-group differences must be built on top of an accumulation of individual-differences (neural features that appear in one group but not the other), and whilst both group-average and trait-level individual difference studies have revealed the engagement of similar brain networks[30,37,38], group-averages cannot explain detailed differences in event representations between pairs of individuals from within the same group.

A question is then posed over how personal signatures of imagined experiences can be identified in fMRI data, and how the information content of brain activity can be interpreted. This would appear to rely on having personalized models of mental imagery that are sufficiently detailed to not only capture differences between events, but also between individuals' idiosyncratic experiences of similar events. Modeling advances[39–45] have led to the ability to predict/decode fMRI activity elicited as third-party sentence-level descriptions of events are read/heard[46–53]. However, such neural decoding models have largely relied upon internet-based approaches to model generic representations of meaning by computationally harvesting semantic representations from massive text repositories, or crowd-sourcing behavioral ratings across large collectives of people. Consequently, the resultant models are built at group-level, and whilst they have high signal-to-noise-ratio when it comes to capturing broad population-level commonalities, they have zero ability to discern individual differences. It is unknown whether current modeling approaches can be tailored to be practically effective at capturing interpersonal differences in imagined experiences. It is also unknown how accurately these current modeling approaches can explain neural activity elicited as individuals vividly imagine situations from their own perspective, as opposed to passively reading third-party sentences[49–53].

We here hypothesize that fMRI activity patterns elicited as people imagine weddings, funerals and eighteen other common scenarios reflect interpersonal differences in the information that is brought to mind. We test this by devising personal models of the imagined experiences to discern participant identity from corresponding fMRI data. In so doing, we provide evidence that: (1) Neuroimaging can quantitatively measure meaningful individual differences in brain activity elicited as people imagine complex events from their own perspective. (2) The information content of brain activation can be predicted using personalized models derived from verbal descriptions and behavioral ratings of the imagined events. We discuss the potential implications this has for both basic and applied science.

## Results

We selected 20 common scenarios (e.g., reading, dancing, restaurant) that we asserted would be common experiences to our study population, whilst also being of a sufficiently broad nature that different people would have different experiences of each (e.g., most people have experienced dancing, whether it was actively or passively, in company or isolation, or enjoyable). We next recruited 30 individuals who were healthy agers, although 26 completed the experiment (17F, 9M, mean ± SD age = 73 ± 7years). Healthy agers were chosen in particular to evaluate the feasibility of the approach for future tests on clinical aging

**1**. 26 participants vividly imagined and then verbally described 20 common scenarios

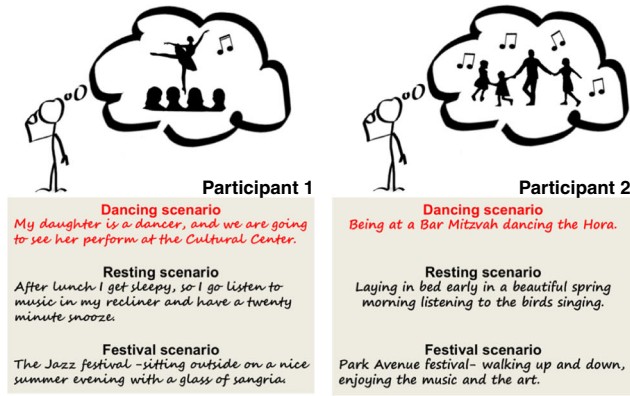

**2**. Participants individually rated their imagined scenarios on 20 **experiential attributes**

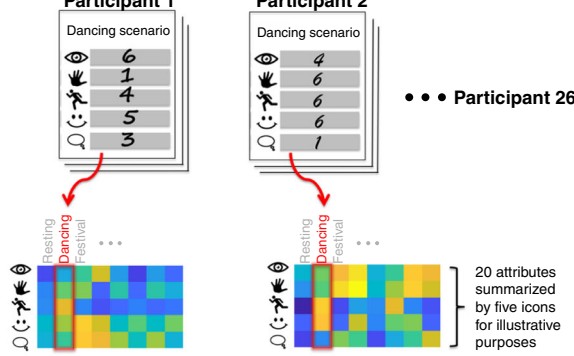

**3**. **Verbal** descriptions were mapped to a text-based distributional semantic model of word meaning

**4**. Participants underwent fMRI as they reimagined the scenarios when prompted by standardized cues

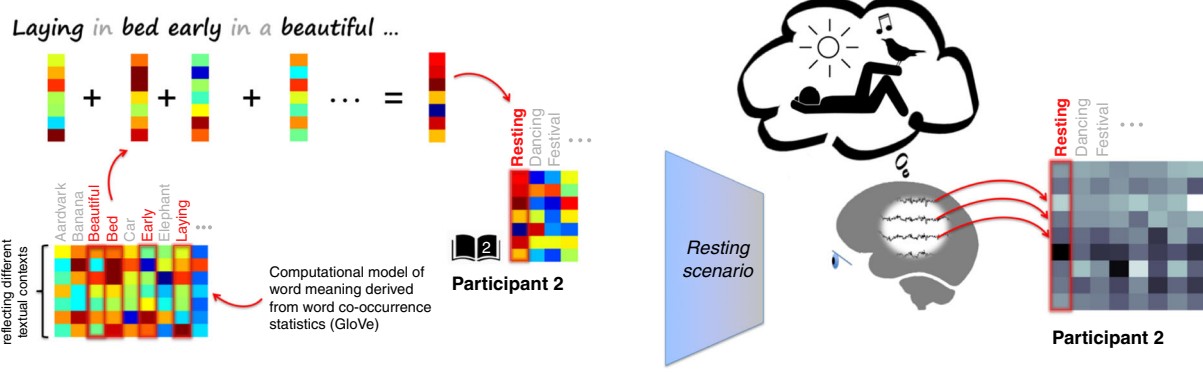

**Fig. 1 Data collection: experimental protocol and construction of personal models of the imagined scenarios.** This diagram summarizes the entire data collection procedure as 4 stages, ranging from the collection and processing of behavioral data to fMRI scanning. The chronological order of stages 1, 2, and 4 reflects the actual order of experimentation. Stage 3 is automated and could in practice take place at any time after stage 1.

populations. In the Discussion we consider what similarities and differences might arise from a comparative study on younger adults, although this remains an open empirical question for the time being.

Each participant attended a visit with an experimenter, who read the different scenarios to the participant one by one (e.g., "A dancing scenario"). On each reading the participant was requested "to vividly imagine themselves in the scenario" placing importance on their (imaginary) perception, action and feelings (Fig. 1). Participants then provided a verbal sentence length description of each imagined scenario. Examples illustrated in Fig. 1 indicate that the descriptions supplied by different participants for the same scenario were at least superficially different. To estimate the extent to which participants could base imagination on real past experiences of the event in question (as opposed to envisioning entirely new fictitious situations) we asked participants to rate this, as well as how vividly they could imagine each scenario. Participants tended to rate scenarios to have happened and to have been vividly imagined (grand mean ± SD ratings across both participants and scenarios were 5.5 ± 1.1, and 5.2 ± 1.2, respectively, on a Likert scale of [0 6]). See Supplementary Information for rating statistics for individual scenarios (Supplementary Fig. 1) and details of the ratings procedure.

**Overview of personalized models of imagined experiences.** To model the person-specific information content of each participant's imagined scenarios, we employed a joint-modeling approach that integrated semantic information that was estimated from participants verbal descriptions of events with non-linguistic sensory, motor, interoceptive, emotional, locational and temporal information that could putatively contribute to episodic recall and simulation (as there is evidence for at least in conceptual representation[53–56]). The linguistic and non-linguistic models were both founded on established approaches that have been extensively used in modeling the semantic representation of words and concepts since refs. [39–45]. Different to previous work that has constructed these models at group-level, we here newly adapted the modeling approaches to capture person-specific scenario representations and newly leveraged them to explain (person-specific) neural activation elicited as participants actively imagined themselves in common scenarios. See also ref. [57] for a language modeling approach to representing individuals and ref. [21] for a personal image-tagging approach to associating autobiographical bodycam photographs with experience).

The linguistic modeling approach, henceforth referred to as the "verbal model", borrowed a popular method from computational linguistics to transform participant's verbal scenario descriptions into quantitative representations reflecting the linguistic meaning of each description. Specifically, we took a state-of-the-art distributional semantic model (GloVe[44]) that approximates words' meaning in terms of the textual contexts that words appear in as measured across a large text corpus. For instance, pyramids and camels are related because they co-occur together

within sentences, and camels and donkeys are related because they occur independently in "riding" sentences. Practically each word is represented as 300-dimensional vector, derived by factorizing a word co-occurrence matrix (vocabulary size is 2.2 million words and co-occurrences were measured across 840 billion tokens from Common Crawl https://commoncrawl.org). We cut out content words (words with intrinsic meaning as opposed to a grammatical function) from participants' verbal descriptions, and mapped each content word to the corresponding GloVe vector. We then combined word-vectors together to model entire scenario descriptions through pointwise addition. Although such additive composition is a naïve strategy that discounts word order and syntax, it has proven practically effective in both computational linguistics[58,59] and in predicting sentence-level brain activation[49–53]. This process yielded a personalized distributional semantic model representation of each participant's verbal descriptions of each scenario for each participant (even though word vectors were drawn from GloVe in each case).

A limitation of verbal descriptions is that they may omit information that is so salient in physical experience that it is assumed to be too obvious to bother expressing. To cite a common example, people seldom feel the need to communicate the color of bananas[60], or that they played tennis in daylight, or watched a movie in the dark, because this is assumed to be common knowledge. To counteract this limitation and potentially build a more comprehensive model of episodic imagery, we estimated the degree that different sensory/motor/cognitive/emotional neural systems would have been engaged if the participant had physically experienced the scenario, and hence might have contributed information to memories and in turn episodic simulation. To this end we constructed an experiential attribute model[45]. Specifically, the attribute model was implemented by having each participant rate each scenario on 20 sensory, motor, cognitive, spatiotemporal and emotional experiential attributes (Likert scale 0–6, see Fig. 1, see "Methods" section and Supplementary Information for specific details). For instance, a regular dancer might be expected to imagine themselves actively engaging at a dancing event, and hence rate it strongly on motor engagement and positive affect. In contrast, an ardent dance-avoider might imagine and rate the opposite. The set of 20 attributes was an abridged selection of a wider set of 65 attributes identified by ref. 45 that when crowd-sourced at group-level, had provided a basis for predicting brain activity elicited as event-related sentences were read[49,52,53]. The 65 attributes were cut down to 20 before experimentation to meet timing constraints (see "Methods" section for further details). Importantly, the attribute model explicitly solicited for a comprehensive rundown of experiential information that participants might not have thought to provide in their verbal reports if left to themselves (even if their reports were open ended in length).

**fMRI experiment overview**. Participants' brain activity was scanned using fMRI as they vividly re-imagined the scenarios (Fig. 1). During fMRI participants imagined their own scenario when a standard written prompt ("A dancing scenario") was visible. To accommodate noisy fMRI measures, the set of 20 scenario prompts were displayed 5 times over (in different random orders), and participants reimagined the same scenarios each time. fMRI data was preprocessed using standard techniques to correct for slice timing and head motion and each participants' fMRI data was spatially normalized to a common anatomical template (MNI). Previous work[14–24,49,52,53] led us to expect that activation patterns within localized brain regions would

comprehensively represent multiple scenarios (in particular the episodic recollection/simulation network[6–8]). Thus, we parcellated fMRI activation the cortex into 90 neuroanatomical regions of interest (ROIs) using Automated Anatomical Labeling[61,62]. We analyzed all regions (rather than a predefined network) to avoid the possibility of overlooking patterns of interest. We estimated the top 100 most informative voxels per ROI using a frequently used voxel stability criteria[46] (see "Methods" section). Comparative results using 50 and 200 voxels yielded a similar pattern of results, and are included in Supplementary Information. The 5 fMRI volumes per scenario were voxel-wise averaged together and vectorized to provide a single 100 dimensional vector per scenario, per ROI, per participant.

**fMRI activation patterns reflect interpersonal differences in scenario representation**. To test our overarching hypothesis that fMRI activation patterns elicited in imagining the scenarios reflected interpersonal differences, we applied Representational Similarity Analysis (RSA)[63] to compare model and fMRI data (Fig. 2 and see "Methods" section). In the case at hand, RSA can be conceptualized as a two-step process that begins by re-representing each of the 20 scenarios as an individual point in a geometric similarity space. This similarity space is common across model and fMRI data to enable their comparison, but the coordinates of the scenarios are defined separately for the model and fMRI data. The second step tests whether the relative geometric arrangement of the scenario coordinates is similar for the model and fMRI data. For instance, if in fMRI similarity space it turns out that the dancing scenario is close to party and wedding, whilst reading is close to writing, and cooking is close to housework, and this same relationship is also seen in model similarity space, then there is evidence that this arrangement is meaningful because the model predicts the representational structure of the fMRI data. Practically, the transformation to similarity space was implemented by computing Pearson inter-scenario correlation matrices separately for model and fMRI data. Model to fMRI similarity space comparisons were implemented by computing Spearman correlation between vectors of values that define the respective fMRI/model correlation matrices (high correlation coefficients reflect a strong relationship, see "Methods" section for more details and a description of the statistical significance testing procedure).

The main interest of the current analysis, was however on detecting meaningful representational structure in fMRI data that is peculiar to each individual, and not a feature of the general population. For instance, if a participant (unusually) reported that they imagined themselves dancing whilst doing their housework then their corresponding model might predict an unusually high similarity between fMRI representations of housework and dancing. To this end, we computed group-average model representations to estimate population-level commonalities in scenario representational similarity structure, and used these as a control in our forthcoming analyses that sought to identify person-specific idiosyncrasies.

All of the primary analyses reported (Fig. 2) incorporated our three personalized data types: (1) fMRI activation (illustrated in gray with the brain icon). (2) The attribute model (illustrated in orange with icons for attributes). (3) The verbal model (illustrated in blue with a book icon). To create a single personal multimodal model representation (illustrated in green with attributes and book icons) we integrated information across the person-specific verbal and attribute models by averaging representations in similarity space (see Fig. 2 and "Methods" section for details, and Supplementary Information for evidence that model integration was beneficial). Group-average models were estimated to serve as

**1.** Re-representation in a common similarity space

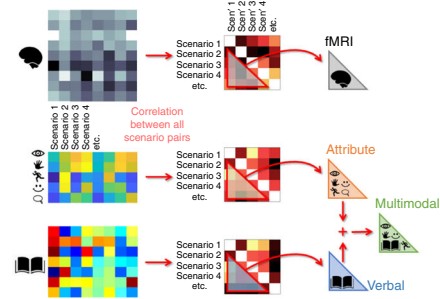

**2.** Building group average representations (G-1)

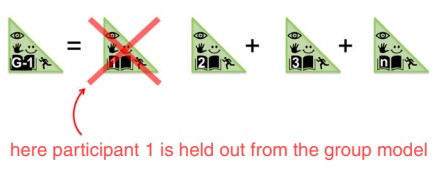

*here participant 1 is held out from the group model*

**3.** Hypothesis: fMRI representations are predicted by personal models over and above group models

**Representational similarity analysis**

Partial corr $\left[ \begin{array}{c} \end{array}, \begin{array}{c} \end{array}, \begin{array}{c} \end{array} \right] > 0$
(Control)

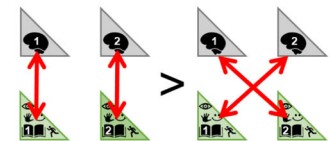

*Matrix triangles were vectorized prior to correlation*

**4.** Hypothesis: individual identity can be decoded by correlating model and brain data

**Decoding individual identity**

Two alternative forced choice
(50% chance)

Hypotheses were tested on all permutations of participants

**Fig. 2 Representational similarity analysis protocol and hypothesis formalization.** The diagram summarizes the computational approach taken in our analyses and illustrates two formalizations of our primary hypothesis. Stage 1 was repeated for each participant. In the diagram fMRI data is represented as a single correlation matrix to simplify visualization, however stages 1 to 4 were repeated on fMRI activation extracted from multiple regions of interest (ROIs). Step 1 illustrates how the three personal scenario representations (fMRI and the verbal and attribute models were transformed into a common space to enable their comparison (stages 3 and 4). Step 2 illustrates how group-average model representations were constructed whilst excluding test participants (G-1). Stages 3 and 4 reflect different approaches to testing the same fundamental hypotheses that fMRI activation patterns reflect meaningful person-specific information that is brought to mind when individuals imagine themselves in different scenarios.

controls in our analyses in a similar fashion by averaging person-specific model representations in similarity space (see Fig. 2 and see "Methods" section for details). Critically a different group-average model was computed for each participant, that excluded the respective participant's model data. So, in a test of participant 1's fMRI, a group-level model (G-1) was built by averaging model similarities across participants 2 to 26 and so on. See "Methods" section for further details. Note, that group-average representations could equally have been built by averaging individuals in model feature space prior to computing similarity matrices, and this approach yields similar results (Supplementary Information).

We first questioned whether fMRI activation patterns in brain regions that reflect group-average model structure (i.e., what we would expect to predict using traditional group-level modeling approaches) also encode person-specific representations. To this end, we initially identified a network of eight brain regions that reflected group-average model structure (excluding the test participant) as is described in detail in the "Methods" section. A complete listing of RSA results for all ROIs is in Supplementary Table 1. The eight ROIs spanned left/right medial parietal cortex, left inferior parietal cortex, left lateral temporal cortex and left mid frontal cortex (the anatomical locations of these ROIs are illustrated in Fig. 3 left). We next tested whether participant-specific models could predict fMRI representational similarities (in the same participant's brain) over and above the corresponding group-average model for the eight ROIs. This was implemented by computing the partial correlation (Spearman) between person-specific model similarity vectors and fMRI similarity vectors, whilst controlling for group-average model similarity vectors. Partial correlation coefficients were r-to-z transformed. Then, for each of the eight ROIs, the set of

coefficients corresponding to the 26 participants were compared to zero using one sample *t*-tests (1-tailed, anticipating positive correlation). The resultant vector of eight *p*-values were corrected for multiple comparisons according to False Discovery Rate (FDR)[64]. Critically, all eight ROIs yielded FDR corrected *p*-values < 0.05. See Fig. 3 right for comprehensive test statistics and Supplementary Table 2 for a complete listing of results on all ROIs. These results provide evidence that person-specific elements of scenario representation were encoded in left/right medial parietal cortex, left inferior parietal cortex, left lateral temporal cortex and left mid frontal cortex. These regions broadly overlap with the core episodic recollection/simulation network, as considered further in the Discussion.

RSA coefficients computed using the personal models in isolation are illustrated in Fig. 4 left. Mean RSA coefficients for the personal models were marginally greater than the group-average models in 7/8 ROIs (Fig. 4, Supplementary Table 1) but there were no statistically significant differences for individual ROIs. The broad similarities in the magnitude of RSA coefficients between the personal and group modeling approaches probably reflect their complementary strengths. Specifically, whilst group models cannot capture the person-specific idiosyncrasies documented above, they have higher signal-to-noise for estimating population-level commonalities. We leave an explicit quantification of the complementary contribution made by the group-models over to future work because it is not the prime focus of the current article.

**Detailed neuroanatomical distribution of person-specific representational structure.** To follow up out previous ROI-based analysis we performed a searchlight analysis[65] to more precisely

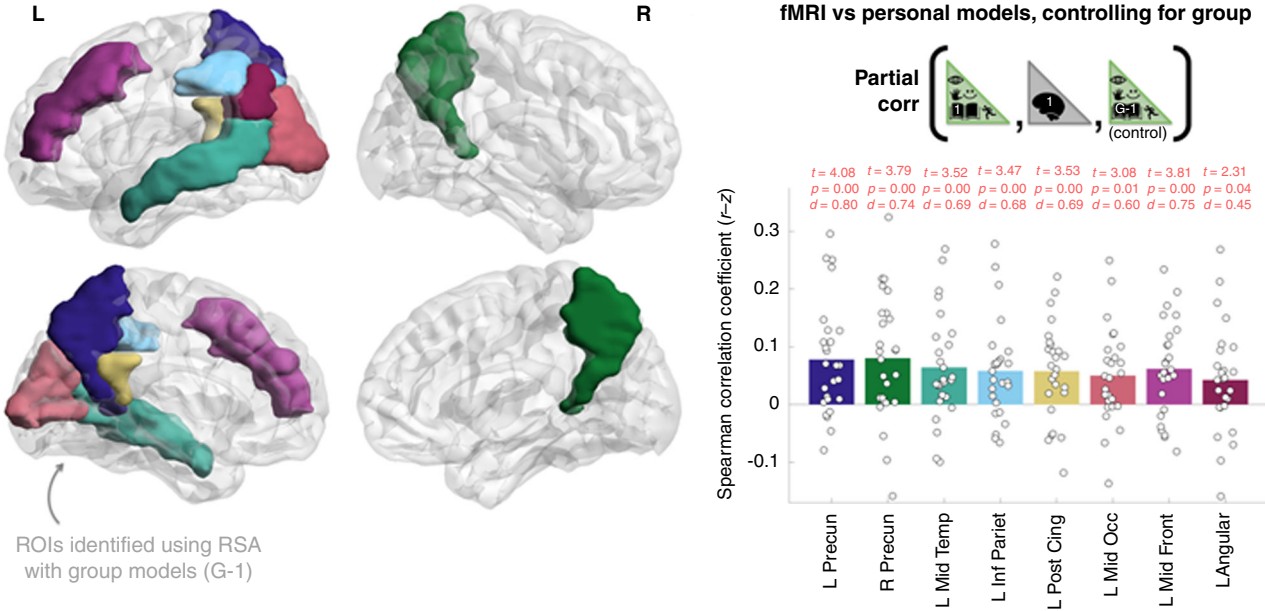

**Fig. 3 fMRI activation patterns elicited in imagining common scenarios reflect person-specific information.** The plot shows how participant-specific scenario models predict the representational similarities of corresponding fMRI data over and above group-average models (that excluded the respective participant in each test). Black circles illustrate RSA coefficients for the 26 participants. Bar heights correspond to the mean across participants. Bar colors correspond to the ROIs illustrated on the brain (left). One sample t-tests tested whether partial RSA coefficients were greater than zero (1-tailed). Cohen's d was computed by dividing the t-statistic by $26^{1/2}$. Exact FDR[64] corrected p-values in the same order as plotted above were: 0.0030, 0.0030 0.0034, 0.0034, 0.0034, 0.0077, 0.0030, 0.0400 (see also Supplementary Table 2). Permutation-based p-values derived from partial RSA tests performed at an individual-level revealed significant outcomes ($p < 0.05$) in the following numbers of participants per ROI (in parentheses): L Precun (7), R Precun (9), L Mid Temp (5), L Inf Pariet (4), L Post Cing (4), L Mid Occ (3), L Mid Front (6), L Angular (4). The cumulative binomial probability of achieving 4 or more outcomes at $p < 0.05$ in an ROI is 0.04. The eight ROIs presented were selected using G-1 data (see "Methods" section, the following figure and Supplementary Table 1 for results using the personal and group-average models separately). Brain illustrations were made using ref. [103]. Source data are provided as a Source Data file.

estimate the neuroanatomical layout of brain regions reflecting person-specific and group-average model structure. To this end we replicated the previous RSA and partial RSA within searchlight ROIs. Searchlight ROIs were cubes of radius 3 voxels (side 7) that were iteratively centered on every location in the brain via the implementation in ref. [66] (as is analogous to shining a searchlight, see "Methods" section). This complemented the previous anatomical ROI-based analysis which was well equipped to detect the presence of person-specific information in fMRI data (because informative voxels were selectively analyzed and weaker assumptions were placed on the shape of patterns of interest, beyond that they could fit into the relatively large anatomical ROIs). However, the previous ROI analysis did not precisely locate person-specific representations.

Results of the searchlight analyses are illustrated in Fig. 5, and the neuroanatomical locations of significant clusters ($p < .05$ FDR[64] corrected) are identified in Supplementary Tables 3–5. Similar to the ROI-based analyses fMRI representations in medial parietal cortex and inferior parietal cortex were identified by the group-average models (Fig. 5 right) and were subsequently found to reflect person-specific information structure in the partial RSA analysis (Fig. 5 left). Different to the ROI-based analysis, prefrontal regions were not detected and inferior parietal cortex was less well represented, which may reflect a lower sensitivity of the searchlight approach (which did not exclude non-informative voxels, and may have been disadvantaged in capturing patterns that did not adequately fit within the searchlight). The searchlight RSA based on person-specific models alone (i.e., when the group-average was not controlled for) revealed a widely distributed network of brain regions that included clusters in dorsal and ventral medial prefrontal cortex, dorsolateral frontal cortex and

anterior temporal cortex (Fig. 5 middle). Importantly, this more neuroanatomically precise estimate of the distribution of regions encoding scenario information echoes the configuration of the core episodic simulation/recollection network[6–8] more precisely than the ROI-based analysis.

**Individual identity can be decoded from brain activity elicited imagining personal experiences.** As a second formalization of our key hypothesis, we explicitly tested how well individual identity could be decoded by matching participants' fMRI activation to their person-specific models. This analysis was repeated on each of the eight anatomical ROIs that had been identified in our earlier analyses. In advance, some degree of success in interpersonal decoding is already entailed from the earlier partial RSA analyses that revealed person-specific neural representations. However, the current analysis is still required to estimate just how accurately pairs of individuals can be distinguished, which would be difficult to estimate otherwise.

The decoding procedure is detailed in "Methods" section. In brief, pairs of test participants were drawn, and we computed RSA between each participant's person-specific model, and both participants' fMRI data (4 coefficients in total). We then tested whether RSA coefficients between participant-specific models and the corresponding fMRI data sets (P1-model vs. P1-fMRI + P2-model vs. P2-fMRI) were greater than the incongruent mismatch (e.g., P1-model vs. P2-fMRI + P2-model vs. P1-fMRI). A decoding accuracy was estimated as the percentage of times that congruent scores were greater than incongruent scores across all 325 participant pairs. If there were no participant-specific relationship an accuracy of 50% was expected.

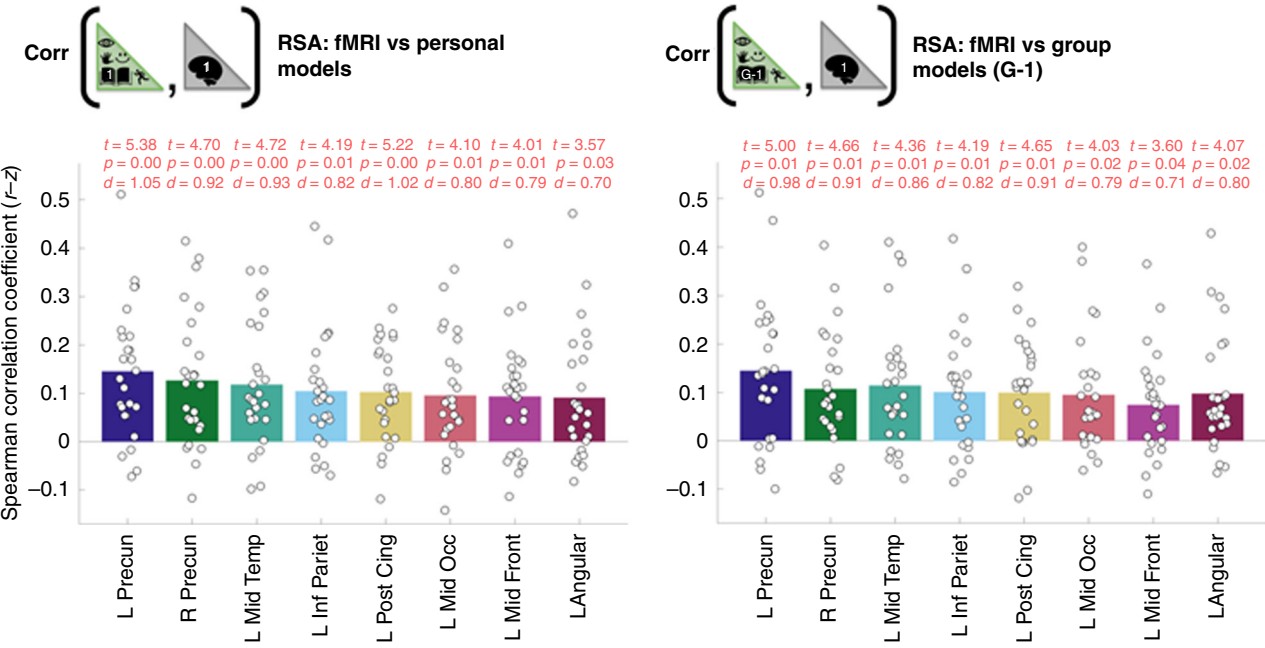

**Fig. 4 RSA coefficients for personal models (capturing idiosyncratic features) and group-average models (high signal-to-noise for group-commonalities) were of broadly similar magnitudes.** Black circles illustrate RSA coefficients for the 26 participants. Bar heights correspond to mean values across participants. t-tests tested whether RSA coefficients were greater than zero. Cohen's d was computed by dividing the t-statistic by $26^{1/2}$. P-values illustrated were FDR-corrected[64] across 90 ROIs. Exact FDR corrected p-values in the same order as plotted above for the personal models were: 0.0024, 0.0047, 0.0047, 0.0116, 0.0024, 0.0124, 0.0138, 0.0327. Exact FDR corrected p-values in the same order as plotted above for the group-average models were: 0.0070, 0.0070, 0.0111, 0.0140, 0.0070, 0.0150, 0.0393, 0.0150. The eight ROIs plotted correspond to regions for which FDR corrected p-values derived using the group-average (G-1) models were less than 0.05 (see also "Methods" section). A detailed listing of results for all ROIs is provided in Supplementary Table 1. Permutation-based p-values derived from performing RSA at an individual-level revealed significant outcomes ($p < 0.05$) in the following numbers of participants per ROI (indicated in parentheses for personal and group-average models, respectively): L Precun (13,14), R Precun (12,8), L Mid Temp (10,11), L Inf Pariet (7,8), L Post Cing (10,9), L Mid Occ (9,8), L Mid Front (8,5), L Angular (8,7). The cumulative binomial probability of achieving 4 or more outcomes at $p < 0.05$ in an ROI is 0.04. Source data are provided as a Source Data file.

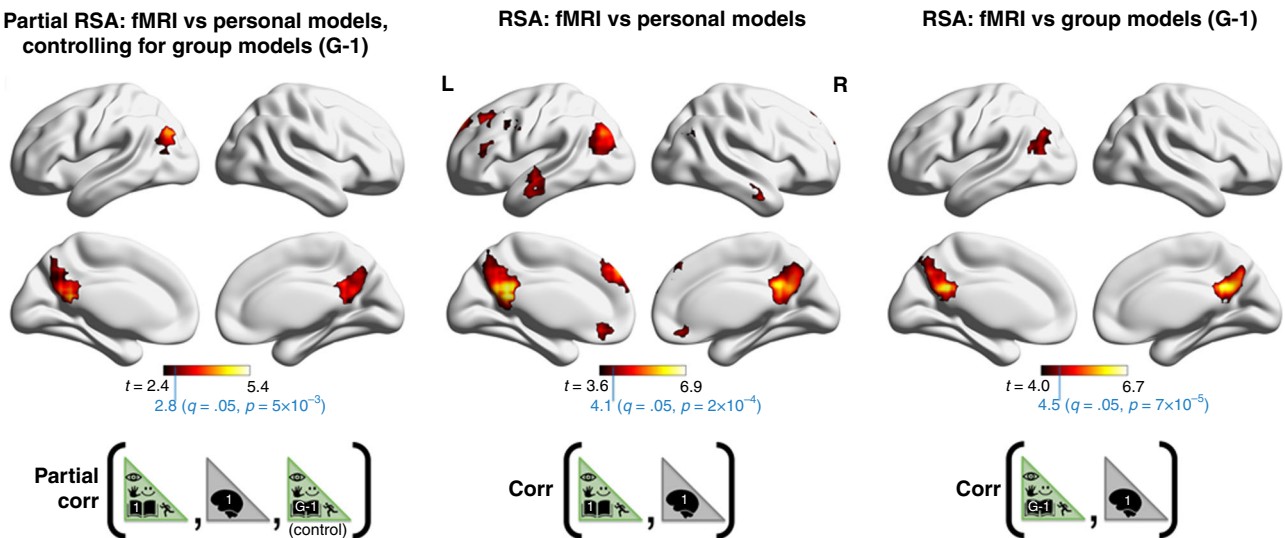

**Fig. 5 Neuroanatomical distribution of person-specific representational structure (RSA-Searchlight).** Computation of the three RSAs illustrated here, as well as hypothesis testing and FDR correction mirrored the protocol of the above ROI-based analyses translated into a searchlight framework. Differently ROI selection was by passing a 3-voxel radius cube throughout the brain (rather than segmenting anatomical atlas regions). The heat maps illustrate t-statistics corresponding to one sample t-tests of the corresponding RSA coefficients against zero. The t-statistics illustrated correspond to p-values that survived an FDR[64] threshold placed at $q = 0.1$ ($q = 0.1$ was used rather than $q = 0.05$ to enhance the visibility of clusters for display purposes). The anatomical makeup of clusters arising from an FDR threshold of $q = 0.05$ are listed in detail in Supplementary Tables 3–5. Brain illustrations were made using ref. [103].

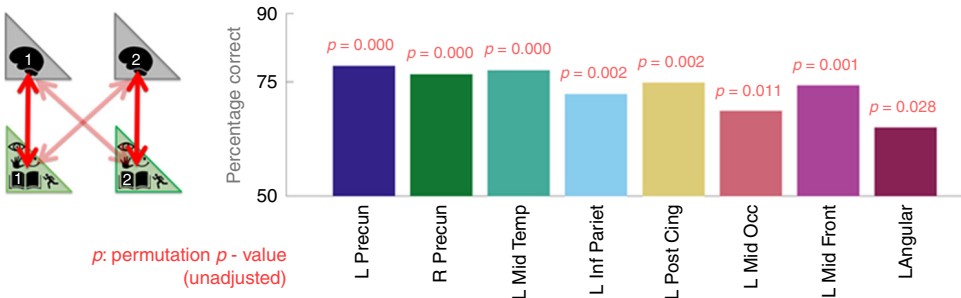

**Fig. 6 Individual identity can be decoded from fMRI activity elicited during the imagination of common scenarios.** Each bar illustrates the percentage of times that participant-specific models better predicted the same participant's fMRI representations than another participants' fMRI data (see main text for details). This test was repeated for each pairwise combination of the 26 participants. The eight ROIs illustrated were identified in our initial ROI-based analysis. Complete results for all ROIs are in Supplementary Table 2. To provide context for the neural-decoding accuracy values, we ran a comparative pairwise decoding analysis in absence of fMRI data based on the verbal and attribute models alone (e.g., P1-verbal vs. P1-attribute and so on). This yielded an accuracy of 83% ($p = 0.0001$). Source data are provided as a Source Data file.

We detected decoding accuracies of around 75% across all eight ROIs (all $p < 0.05$, see Fig. 6 for precise percentages and permutation test-based $p$-values, and Supplementary Table 2 for a complete listing of results for all ROIs). Critically, these results provide evidence that individual identity can be decoded from fMRI activity elicited during the imagination of common scenarios. They also provide an initial estimate of how accurately pairs of individuals can be distinguished based on fMRI data.

**Supplementary analysis overview: results are robust to perturbation of parameters.** In light of the variability in hypothesis testing outcomes that have been observed to arise out of differences in choices of analytic procedure[67], we conducted a battery of analyses to probe the robustness of the current results to changes in parameterization.

As concerns the anatomical ROI-based RSA we present evidence in Supplementary Fig. 2 that both the verbal and attribute models helped contribute to explaining person-specific fMRI representations (and therefore that multimodal model integration was warranted). In Supplementary Figs. 3 and 4 we demonstrate that the verbal and attribute models both predicted person-specific fMRI representational structure when applied in isolation, and therefore that even though multimodal model combination proved to be beneficial, it was not essential to detect interpersonal differences. In Supplementary Fig. 5 we replicate our main findings when group-average models were differently built by averaging individuals' model representations in model feature space as opposed to similarity space (thus the averaging method was not critical). To demonstrate that results were not tied to the precise number of voxels used in analysis, we replicated similar outcomes when analyzing 50 or 200 informative voxels per ROI (Supplementary Figs. 6 and 7). For completeness, we also include results derived using RSA on generic GloVe representations of the stimulus prompts (GloVe vectors did correlate with the eight ROIs, but cannot account for interpersonal differences, see Supplementary Fig. 8). Lastly, we repeated the RSA when left handers' brain hemispheres were flipped to counteract putative left lateralized language effects, which made little difference (Supplementary Fig. 9).

For the identity decoding analysis (Fig. 6) we generated broadly similar conclusions when analyses were repeated using the verbal and attribute models in isolation (Supplementary Fig. 10) on either 50 or 200 informative voxels per ROI (Supplementary Figs. 11 and 12), and when analyses excluded participants with weak model-to-brain correlations (Supplementary Fig. 13), and when analyses were performed within female/male participant subgroups (Supplementary Figs. 14 and 15).

## Discussion

This study has decoded individual identity from brain activity elicited during the imagination of a set of common situations. Importantly, this has provided evidence that fMRI can measure brain activity with sufficient signal to quantify meaningful interpersonal differences in the neural representation of complex imagined events. In parallel, we have shown that the representational structure of person-specific fMRI signal can be predicted using the current personalized verbal and attribute models. This constitutes a critical advance beyond the one-size-fits-all approach that has typically been applied to predict fMRI data based on group-level models derived from internet data[46–53]. Whilst such group-level models have high signal-to-noise-ratio for capturing population-level commonalities, they cannot distinguish individuals, and they cannot predict person-specific variance across fMRI representations of multiple stimuli as we have done here.

This study revealed that fMRI activation patterns in left/right medial parietal cortex, left temporoparietal junction, left dorsolateral prefrontal cortex, ventral and medial prefrontal cortex, and anterior temporal lobe reflected the representational structure of individual participants' personal descriptions/ratings of their imagined scenarios. This distribution of regions closely resembles the layout of the established core episodic recollection and simulation network[6–8]. Our results are probably best considered to reflect a mixture of recollection and simulation. However, the two are practically difficult to disambiguate here and even some of the scenarios that participants reported as being recollections of real events may have reflected a conflation of information across multiple related episodes[68] (as might be particularly prevalent in our more routine experimental scenarios, such as driving, housework and cooking).

In particular, we found that medial parietal cortex (MPC: precuneus and posterior cingulate) encoded person-specific representations strongly. MPC is routinely activated in episodic recollection/simulation and also in the perception of scenes and space[10–13,69]. MPC is believed to play a role in segmenting events from continuous experience and encoding abstract cross modal event representations during recall[16,17]. Other recent work has revealed that imagined places, contexts and people can all be discriminated from fMRI activation within MPC[19–23] and that MPC — temporal lobe connectivity reflects the content of ongoing thought[70] and memory recall[22]. Our study adds to this by revealing that MPC representations of complex imagined events are person-specific and can be predicted using the current verbal/attribute models.

The left temporoparietal junction (LTPJ: posterior mid temporal cortex, inferior parietal cortex, angular gyrus and mid

occipital gyrus) also strongly encoded person-specific information. Besides sharing a role in event segmentation and simulation[16,17], LTPJ has been linked to the so-called autonoetic[71] conscious subjective experience associated with remembering episodes from a first-person perspective[13,72], as well as bodily self-consciousness[73]. Indeed, a novelty of the current approach was the initiative to model imagined scenarios from participants' first-person perspectives. However, whilst this approach proved successful, future work will be necessary to evaluate what role perspective played here. For instance, our participants did not additionally describe/rate scenarios from a third-person perspective, so we cannot be sure that such a model would not have worked just as well. We hope to elaborate on this in the future.

Left Dorsolateral prefrontal cortex (DLPFC: mid frontal) also reflected individual differences. DLPFC is typically associated with cognitive control processes that regulate thought and actions according to task goals/personal interests[38,74,75]. Thus, DLPFC is likely to have helped regulate the activation and suppression of scenario memories on cue, possibly by deactivating memory codes in the hippocampus[76,77]. It is interesting (though not unexpected[20]) that DLPFC would encode detailed scenario representations given its putative role in cognitive control. One possibility is that these representations were read from other brain regions to help inform task prioritization. Alternatively, representations could be indirect, reflecting the differential engagement of control processes invoked in the recall/simulation of different modalities of information. Future work exploring the dynamics of information flow between regions may help to clarify this.

The hippocampus, which is a key hub of episodic memory[13] and simulation[2], was conspicuous by its absence from our results. The hippocampus has been described in terms of providing a spatial scaffolding for episodic memory that indexes the neocortical components that code perceptual, emotional, and conceptual content of experience and provide a sense of autonoetic consciousness[13]. In light of this, it is reasonable to suppose that the current attribute/verbal models may better predict the reconstructed neocortical perceptual/emotional/conceptual content than the hippocampal scaffolding. This said, close inspection of our results (Supplementary Tables 1 and 2) reveals that hippocampal (and parahippocampal) ROIs did significantly reflect personal scenario representations prior to correction for multiple comparisons.

In addition, our searchlight analyses identified clusters in: (1) Ventromedial prefrontal cortex, which has recently been found to encode the identity of known people, places and whether they were liked when fictitious meetings are imagined[7], and more generally is thought to play a role in encoding self-reference[78], emotional information[79], and remote (years-old) episodic memories[13,15]. (2) Dorsomedial prefrontal cortex, which has been associated with inferring traits of other people[78], and has recently been shown to encode collective memories reflecting sociocultural group membership[24]. (3) The anterior temporal lobe, which is thought to be a key semantic memory hub[80].

It is noteworthy that beyond the anterior temporal lobe, most, if not all of the regions uncovered in our analysis, and more generally most of the core episodic recollection/simulation network[6–8] overlaps with the brain's semantic system[79]. Whilst it has always been acknowledged that episodic and semantic memory share a close relationship, they have classically been distinguished on phenomenological grounds[70,81,82]: Episodic memories are characterized by an awareness of the self, situated somewhere at a unique time and place, whereas semantic memories reflect the sort of general knowledge found in an encyclopedia that lacks a personal space-time context. Since their initial distinction[70], the boundaries between the two have become increasingly fuzzy both in terms of theory and also their neural correlates[83,84]. One proposal is that this is because episodic memories bind together familiar semantic concepts (e.g., people, items, places) within an episode-specific sensory and spatiotemporal context[83]. Indeed, this is roughly what our modeling approach has recreated in integrating generic word-level semantic vector representations with scenario-level experiential attributes.

The broader relevance of the success of the current approach is to ground the proposal that it is now practical to study the neural correlates of both episodic and semantic memories within the same unified predictive modeling framework. The verbal and attribute models deployed here were constructed from much the same parts that we[49,52,53] (and others[46–48,50,51]) have used to predict semantic brain activity elicited in language comprehension. In particular, in a separate study, performed on different participants, we have shown how word-level attribute vectors can be integrated with GloVe vectors (both group-level) to predict fMRI activity elicited in (third-party) sentence comprehension. This revealed a "semantic network" of regions that is broadly similar to the episodic simulation network of this study. A difference was that semantic predictions were relatively more accurate in temporal and inferior frontal regions[49,52,53] than in medial parietal cortex (contrary to the case here). However, the key point is that the same modeling basis can be adapted to predict brain activity in the phenomenologically different settings of episodic simulation and language comprehension. A potential advantage of such a unified approach would be to enable features that are believed to differentiate episodic/semantic memory (time/place/autonetic awareness) to be explicitly encoded in a predictive model and evaluated on their ability to predict brain activity elicited in episodic/semantically oriented task contexts.

Indeed, the notion of deploying unified episodic/semantic models could be relevant to other cognitive domains. It has been suggested[84] that episodic and semantic memory systems are anchored on the default network[85]. The default network encompasses multiple interwoven systems[86] that are active in ongoing thoughts[29] such as remembering, envisioning the future and making social inferences. Studies attempting to characterize ongoing thought have typically applied "multidimensional experience sampling", where participants repeatedly rate the nature of their current thoughts on multiple dimensions[29] (e.g., whether thought is detailed, in images or verbal, related to the past or future…). Not only do within-participant thought samples covary with concurrent default network activity[30,38], but individual differences in resting-state network activity profiles predict trait-level individual differences in thought profiles sampled outside the scanner at a later occasion[70], as well as other psychometric traits[37]. Thus, accepting that ongoing thought includes episodic recollection/simulation, we hypothesize that individual differences in the current fMRI measures/models will to an extent reflect individual differences in thought profiles sampled in the wild, and that both measures will combine to predict individual differences in personality and other stable psychometric traits. We also hypothesize that the current neural measures and verbal/attribute models could provide new ways to predict individual-differences in ongoing thought at a state-level, tracking how the "meaning" of internal monologues[86,87] and/or the "experience" of daydreaming unfolds over time, and possibly even may forecast how novel future events will be experienced (see also refs. [56,88,89] for image-based computational models of visual imagery).

Beyond basic science, we contend that the current methods could contribute to clinical studies where the ability to detect individual differences is essential at various levels. This could be in helping to characterize and diagnose disorders associated with episodic memory and imagery deficits e.g., Alzheimer's

disease[90,91], schizophrenia[92], and depression[93], and perhaps even personalizing treatments and predicting outcomes (e.g., refs. [94,95]). This study would suggest the current approach is feasible for use in healthy agers at least. How the current results extend to younger adults remains untested. We naturally expect interpersonal differences to also be discernable in younger adults' fMRI data (who are less prone to neural atrophy[96,97]). However, we also anticipate there will be systematic differences associated with socioecological factors, life experience, and neurophysiology[97]. In particular we hypothesize that younger adults' fMRI data will be characterized by the presence of relatively more episodic features, based on behavioral studies showing that older adults' verbal descriptions of past[98] and future-hypothetical[99] scenarios contain more semantic details (not specific in time and place), and fewer episodic details (specific in time and place).

In conclusion, this study has both revealed and predicted meaningful interpersonal differences in fMRI activity elicited as personal experiences were imagined. Firstly, we hope that our approach will help provide a unified modeling foundation for future studies seeking to predict interpersonal differences in the neural correlates of episodic and semantic memory as well as other related cognitive domains. Secondly, we hope that our approach will improve the predictive power of future fMRI studies by demonstrating how person-specific between-stimulus variation can be modeled and explained in terms of model features/scenarios. Finally, we hope that that this work presents a step towards future methods that enable detailed information elicited in mental imagery to be read directly from the brain.

## Methods
The study was approved by the University of Rochester research subject review board (RSRB00067540). All participants were required to understand the experimental procedure and give their consent by signing an informed consent form.

**Participants**. Thirty healthy older adults with normal cognition, indexed by Telephone Interview for Cognitive Status (TICS^TM) ≥ 31, were recruited for the study. Participants had adequate visual (normal or corrected to normal vision) and auditory acuity for testing, were English-speaking, and community dwelling. Of the 30 participants, 26 yielded viable fMRI data: 2 participants failed to attend fMRI and 2 attended but failed to complete the experiment. Of the 26 participants included in analysis, 5 were left handed and 17 were female. Mean ± SD age was 73 ± 7 years old.

**Scenario stimuli**. We identified 20 common scenarios that we anticipated: (1) would have diverse experiential characteristics e.g., whether they are actively or passively experienced, social or asocial, indoors or outdoors etc; and (2) that different participants would have different experiences of e.g., in their degree of activity and social engagement, in venue and so on. This was with the goal of generating sufficient statistical structure in both model and neural data, such that both scenarios and participants would be distinguishable. The number of scenarios was restricted to 20 to meet experimental and scanning time constraints (see also the Experiential attribute model section below). Participants were asked to "vividly imagine themselves in each scenario" prior to providing a brief "sentence-length" description of that scenario. They were later requested to re-imagine the same scenario whilst undergoing fMRI. Scenarios were first read to participants, and later displayed during fMRI in the following form: "A X Scenario", or "An X Scenario" where X is a placeholder for: resting, reading, writing, bathing, cooking, housework, exercising, internet, telephoning, driving, shopping, movie, museum, restaurant, barbecue, party, dancing, wedding, funeral, festival.

**Experiential attribute model**. To characterize participants' imagined experience of the different scenarios we identified 20 experiential attributes. These were selected from a broader set of 65 attributes, which had recently been introduced to model semantic representation using peoples' ratings of their sensory/motor/affective/ cognitive experiences with words and their referents[45]. e.g., "Please rate the degree to which your restaurant scenario involves human speech sounds". The 65 attributes were cut down to 20 to meet experimental constraints such that participants could read, imagine, describe and rate the 20 scenarios comfortably within a 2-h time frame. The 20 attributes were selected to broadly span the different domains of experience that were originally identified by Ref. [45]. However, the final choice was ultimately based on the current authors' intuitions of which attributes were most relevant for the current scenarios, as guided by prior experience using the 65

attributes to model fMRI data[49,51,52]. The attributes are listed as follows: bright, color, motion, touch, audition, music, speech, taste, head, upperlimb, lowerlimb, body, path, landmark, time, social, communication, cognition, pleasant, unpleasant. On rating each scenario, the participant was reminded of their personal verbal description of the scenario. A standard description of each attribute was provided to guide ratings, along with examples of words that would receive high and low scores (selected from the original templates introduced by ref. [45], and listed in detail in Supplementary Table 6). Each attribute rating was normalized within each participant, by subtracting the mean rating for the attribute (across 20 scenarios), and dividing by the standard deviation.

**Verbal model**. To quantitatively model participants' verbal descriptions of their imagined scenarios, we applied a state-of-the-art and freely downloadable distributional semantic model of word-level meaning (GloVe[44]). GloVe represents words as 300 dimensional vectors (see main text in "Results" section for more details). We cut out content words from participants' verbal descriptions of scenarios, and mapped each content word to the corresponding GloVe vector. We then combined word-vectors together to model entire scenario descriptions through pointwise addition.

**fMRI stimulus presentation**. Stimuli were presented on a screen in black Arial font (size 50) on a gray background that participants viewed while in the scanner. fMRI was scanned during a single uninterrupted session in which the 20 scenario stimuli were presented five times over (five runs). Scenario order was randomized within each run. Scenario stimulus prompts (e.g., "A dancing scenario") remained on screen for 7.5 s (3 TRs). The participants had been instructed to re-imagine themselves in the given scenario only when the stimulus prompt was on screen and clear their mind after it disappeared. There was a 7.5 s (3 TR) delay prior to the next scenario presentation, during which time a fixation cross was displayed. Runs were separated by a 15 s interval, in which a second by second countdown was displayed (e.g., "Starting run 2 in 13 s"), which was followed by 7.5 s of fixation cross preceding the first stimulus of the run. The 26 participants who successfully completed scanning all reported that they had been able to imagine the scenarios on prompt.

**MRI data collection parameters**. Imaging data were collected at the Rochester Center for Brain Imaging using a 3 T Siemens Prisma scanner (Erlangen, Germany) equipped with a 32-channel receive-only head coil. The fMRI scan began with a MPRAGE scan (TR/TE = 1400/2344 ms, TI = 702 ms, Flip Angle = 8°, FOV = 256 mm, matrix = 256 × 256, 192 sagittal slices, slice thickness = 1 mm, voxel size $1 \times 1 \times 1$ mm$^3$). fMRI data were collected using a gradient echo-planar imaging (EPI) sequence (TR/TE = 2500 ms/30 ms, Flip Angle = 85°, FOV = 256 mm, 90 axial slices, slice thickness = 2 mm, voxel size $2 \times 2 \times 2$ mm$^3$, number of volumes = 639).

**MRI preprocessing**. SPM 12 was used to preprocess participants' structural and functional MRI data. Structural scans were co-registered and warped to a common anatomical template in MNI space. The first 10 functional scans during which time steady-state equilibrium was achieved were discarded from the fMRI data. This resulted in the deletion of fMRI volumes associated with the very first (randomly selected) scenario prompt. Thus, a single scenario per participant was represented by 4 rather than 5 fMRI replicates, but otherwise the scenario was treated the same in analyses. The remaining scans were slice-time corrected, motion-corrected, co-registered to their normalized structural images, and then warped to MNI space by applying the same transformation which normalized their structural image. fMRI data was resampled at $3 \times 3 \times 3$ mm$^3$.

Six head motion parameters (translation on x, y, z axes, and yaw, pitch and roll, see Supplementary Fig. 18) and linear trend were voxel-wise regressed out from the fMRI data within each of the 5 runs. Specifically, prior to regression, each vector of voxel activation, and each vector of the nuisance parameters (motion/trend) was z-scored within each run by subtracting the vector-wise mean and dividing by the vector-wise standard deviation. A separate multiple regression was computed to predict voxel activation from the nuisance parameters. Finally, the residuals arising from the regression for each voxel were taken forward to use in computing the scenario representations that would form the basis of our subsequent analyses. Individual scenarios were represented by voxel-wise averaging across the four fMRI volumes (of residuals) spanning the period 5 to 15 s after stimulus onset (at 15 s the next stimulus prompt was displayed). The point of onset at 5 s was selected because the hemodynamic response associated with imagination was likely to peak at the very earliest at 5 s post stimulus presentation (the visual response to the written stimulus would be expected to peak at around 5 s, with recall/imagination following later). The interval of 4 volumes were measured to allow for between-participant and between-scenario differences in hemodynamic response profiles. Such hemodynamic differences could arise from a mixture of differences in neural response latencies associated with consciously retrieving scenarios from memory and bringing them to mind, reliving the multisensory experience of the simulated scenario (which could play out for different durations for different scenarios/people), and subsequently suppressing the imagined scenario.

Besides having observed averaging activation to be an effective strategy in other researchers' studies of active thought[35,46] and in our own work on mental simulation/active thought[56,88,100], we chose the averaging approach in favor of a canonical HRF-based approach because we were not confident that some of the assumptions made by the canonical approach would hold for the current task (which relies on participants to consciously coordinate their recall/imagination, as opposed to say testing for automatic perceptual responses). In particular, the canonical HRF approach would both assume and predict precisely the same neural response profile (including peak response latency and response duration) for each participant and each scenario. This would be modeled by convolving a time series marking the stimulus display periods with a fixed canonical hemodynamic response function. We were initially cautious about this assumption based on our own experience as dedicated participants undertaking different mental simulation studies of a similar ilk, where we have found it difficult to rigorously synchronize our conscious imagination to stimulus onsets and disappearances. To follow up these hunches we performed a preliminary investigation into the degree of interpersonal variation in hemodynamic responses for the current data.

Results of this investigation are included in Supplementary Figs. 19–21. Qualitative inspection suggests that there was indeed substantial interpersonal variability in both the latency and duration of participants' hemodynamic response profiles. For example, some participants appear to have responded rapidly in imagining scenarios and also appear to have cleared their minds relatively sharply. Other participants appear to have responded more slowly and to have kept mental simulations going on for longer. Thus, modeling each participants' activation using the same HRF would have captured some participants' neural response profiles well and others poorly. As the goal of this study was to capture interpersonal variation, we opted for the averaging approach in attempt to avoid biasing our results to reflect only the subset of participants who exhibit canonical responses. We do plan to pursue the nature of these interpersonal (and potentially inter-scenario) differences in neural response dynamics in future work. However, for the time being the current averaging approach provides a practically effective solution to the current problem that lacks the assumption of common peak response latencies and durations across participants. For the future, we note that there may be benefits to including the temporal derivative of the HRF in computation, as was suggested by a reviewer.

**fMRI Voxel selection (within regions of interest)**. Because not all fMRI voxels contain informative signal, we estimated which ones were likely to be informative using a commonly used strategy introduced by ref. [46]. This identifies stable voxels that had a similar response pattern across each run of 20 scenarios. For each participant, and separately for each voxel, we took each pair of runs, and computed the inter-run Pearson correlation in activation across the 20 scenarios. This resulted in 10 correlation coefficients per voxel (derived through intercorrelating the five runs) that were r-to-z transformed. A single stability metric for each voxel was derived by taking the mean of the 10 coefficients.

We then parcellated fMRI data into 90 neuroanatomical ROIs using Automated Anatomical Labeling[61,62]. We repeated all of our main analyses on the 100 most stable voxels (marked by high positive coefficients) within each ROI, or all voxels in the ROI if there were fewer than the desired number. Stable voxels were extracted from each ROI, and vectorized. A single representation of each scenario was computed by voxel-wise averaging activation across the 5 scenario replicates. Activation in each voxel was then normalized by subtracting the mean and dividing by the standard deviation of that voxel's activation. To test the robustness of our results to this particular parameterization, we repeated analyses on the 50 and 200 most stable voxels per ROI and obtained similar results, which are reported in Supplementary Information.

**Transformation of personal models and fMRI data to representational similarity space**. To test for a relationship between model and fMRI scenario representations we applied Representational Similarity Analysis (RSA)[63]. To set this up, fMRI data and models were re-represented in similarity space. For each of the 26 participants, inter-scenario Pearson correlation matrices were computed separately for the verbal and attribute models (yielding two 20*20 matrices). All correlation coefficients were r-to-z transformed (arctanh). The procedure was repeated on each participants' fMRI data, and performed separately on each ROI to generate a total of 90 correlation matrices per participant. The below diagonal matrix triangle (similarity triangle) was segmented from all (model and fMRI) correlation matrices (which are symmetric on the diagonal). Each similarity triangle was then vectorized to form a similarity vector with 190 entries.

**Multimodal similarity model construction**. To construct a multimodal model representation for each participant, their (personal) verbal and attribute similarity vectors were normalized by z-scoring, i.e., subtracting the mean value (of the 190 entries) from each entry and dividing each entry by the standard deviation (across the 190 entries). A multimodal representation was computed for each participant by pointwise summation of the verbal and attribute similarity vectors.

**Group-average (G-1) similarity model construction**. To construct group-average model representations that estimate population-level commonalities in scenario representation, we pointwise averaged model similarity triangles corresponding to the group members. Critically a different group-average model was built for each test participant which excluded that test participant's data. So, in a test of participant 1's fMRI, a group-level model (G-1) was built by averaging model similarity triangles for participants 2 to 26 and so on. Note that group-level representations could equally have been built by averaging individuals in model feature space prior to computing similarity matrices, and this approach yields a similar outcome (Supplementary Fig. 3).

**Representational similarity analyses correlating fMRI data against model data**. To test for a relationship between model and brain data, Spearman's correlation was computed between model and brain similarity vectors. The arising correlation coefficients provided a quantitative measure of the strength of match between fMRI and model representations. The statistical significance of RSA correlations was computed at an individual-level using a standard permutation testing procedure[63]. Scenario order was randomly shuffled, rows and columns of the fMRI correlation matrix (but not model matrix) were rearranged following the shuffled order, the lower matrix triangles of both fMRI and model matrices were extracted, vectorized and then compared to each other using Spearman correlation. This was repeated 1000 times with different shuffles, and the 1000 resulting coefficients were used to build a NULL distribution. Statistical significance was estimated as the fraction of randomly shuffled correlation coefficients that were greater than or equal to the unshuffled correlation coefficient. This produced a p-value for each individual participant. For instance, the number of participants with permutation p-values < 0.05 is listed in relevant figure captions and indicated by i* in Supplementary Information plots.

**ROI selection based on other participants' data**. Because many ROIs were unlikely to represent information associated with the imagined scenarios (i.e., ROIs that are outside the recollection/simulation network), we identified a subset of regions on which to focus our primary individual differences analysis. Our initial interest here was in establishing whether ROIs that represent common group-average scenario structure are further characterized by person-specific idiosyncrasies. This was based on the natural assertion that despite person-specific idiosyncrasies, different people would imagine different scenarios in a broadly similar way (see also ref. [16]). We first tested this assertion of cross-participant commonality by performing RSA to compare person-specific multimodal models between each pair of participants. In total this test was performed on 325 pairwise combinations of the 26 individuals. This yielded a mean ± SD Spearman's correlation coefficient of 0.33 ± 0.11, which when r-to-z-transformed was significantly greater than zero ($t = 52.8$, $p < 1e{-}128$). This supported our assertion that there were indeed strong commonalities between scenario representation between individuals.

Then, we isolated ROIs that reflected group-average representations across participants to later test for person-specific information. We ran RSA between each ROI and group-average models (that excluded the test participant) for each test participant. This produced an RSA correlation coefficient for each ROI and each participant. All correlation coefficients were r-to-z transformed. Then, for each separate ROI, the set of coefficients corresponding to the 26 participants were compared to zero using one sample t-tests (1-tailed, anticipating positive correlation). This yielded a vector of 90 p-values. The 90 p-values were corrected for multiple comparisons according to False Discovery Rate[64] (FDR). FDR corrected p-values corresponding to eight ROIs were under the conventional $p = 0.05$ threshold. These were: Precuneus_L, Precuneus_R, Cingulum_Post_L, Occipital_Mid_L, Parietal_Inf_L, Temporal_Mid_L, Frontal_Mid_L, and Angular_L. We considered these eight ROIs provisionally as candidates to take forward to the next person-specific test.

However, before continuing forward, we tested whether this selection of ROIs was reasonable from a different perspective. The above ROI selection approach risks overlooking ROIs that might solely correlate with person-specific models (and not the group-average models) and besides, also carries a risk of emphasizing effects associated with the group-average models (if group-average models happen to fit fMRI noise that is not subsequently captured by the personal models). Therefore, we further tested ROI selection based on comparisons between other participants' fMRI data and their personal models. Specifically, to select ROIs for an individual participant (e.g., P1), we computed RSA between every other participant's (e.g., P2 to P26) fMRI data and their corresponding personal models. This was repeated for each ROI. RSA correlation coefficients were r to z transformed. For each ROI, the set of 25 (n-1) coefficients were compared to zero using one sample t-tests (one-tailed, anticipating positive correlation). This yielded a vector of 90 p-values. The 90 p-values were FDR corrected and ROIs with adjusted p-values < 0.05 were selected for that participant (e.g., P1). Repeating this procedure across all participants resulted in 7/8 of the previously selected ROIs being identified in every one of the 26 participants. The left-out ROI was Angular_L which was selected in 20/26 participants. In addition, Frontal_Med_Orb_L, Frontal_Sup_L were identified for every single participant, alongside a mixture of other ROIs that differed between participants. To simplify the interpretation and visualization of results we performed our forthcoming analyses on the original fixed set of eight ROIs (rather than a different subset of ROIs for each individual). A comprehensive listing of test statistics for all 90 ROIs

is in Supplementary Table 1. Frontal_Med_Orb_L, Frontal_Sup_L are later identified in our searchlight analysis.

**Partial RSA between fMRI and personal models controlling for group-average representations**. To test for idiosyncratic person-specific correlations between personal models and the corresponding participant's fMRI data, we applied Spearman partial correlation. Specifically, partial correlation was computed between each participant's fMRI similarity vector and their personal model similarity vector whilst controlling for the corresponding group-average similarity vector (where the group-average excluded the respective participant). To estimate the statistical significance of partial RSA correlation coefficients at a single individual-level we applied a permutation testing procedure[101,102]. The procedure differs from case of the standard RSA correlation between fMRI and a single model, where it is sufficient to shuffle the fMRI similarity vector as detailed in the previous section. The difference arises because in the case of partial correlation (and multiple regression) shuffling the fMRI vector disrupts its relationship with both the personal similarity vector and the control (group-average) similarity vector, rather than just with the person-specific vector alone (which is desired). To this end, as described in refs. [101,102] the relationship between the control vector and fMRI vector can be regressed out from the fMRI vector to leave a vector of residuals. The residuals can then be shuffled, and the control effect that was just previously regressed out can then be added back onto the shuffled residual vector prior to computing partial correlation.

In more detail, each permutation was evaluated as follows: (A) All values in fMRI, personal and group-average similarity vectors were ranked (as is standard in Spearman correlation). (B) The ranked fMRI similarity vector from A was regressed on the ranked group-average similarity vector. (C) The residuals arising from B were permuted. To accomplish this, the vector of residuals was entered back into the appropriate (triangular) positions of a 20 by 20 (inter-scenario) matrix. The rows and columns of the residual matrix were then shuffled according to a random order (as in RSA). The below diagonal matrix triangle of shuffled residuals was extracted from the matrix and vectorized. (D) The group-average similarity vector was projected back into fMRI similarity space by multiplication with the beta-weights arising from the regression in B. (E) The result from D was added on to the shuffled residual vector from C. (F) The partial correlation was computed between the result from E and the ranked personal model similarity vector, controlling for the ranked group group-average similarity vector.

Steps A to F were repeated 1000 times with different shuffles (in C), and the 1000 resulting coefficients were used to build a NULL distribution. Statistical significance was estimated as the fraction of randomly shuffled correlation coefficients that were greater than or equal to the unshuffled correlation coefficient. The number of participants with permutation $p$-values < 0.05 is listed in relevant figure captions and indicated by $i*$ in Supplementary Information plots.

**Searchlight RSA/partial RSA analog of anatomical ROI-based analysis**. To build a more precise picture of the neuroanatomical distribution of scenario representations, we replicated the previous ROI analysis under a searchlight framework[65] by passing a cube of voxels (radius 3 voxels, side 7 voxels, mean ± SD number of gray matter voxels per cube ≈257 ± 56) throughout the brain (using the implementation presented in ref. [66]). RSA was computed within each searchlight ROI of each participant's brain, using group-average models and person-specific models. RSA coefficients were r to z transformed (arctanh), and assigned to the center of the searchlight ROI. Voxels that were common to all participants were segmented, and one sample $t$-tests were applied at each voxel location to test whether RSA coefficients were significantly greater than zero. FDR[64] correction was applied to $p$-values corresponding to all voxels (separately for the person-specific test and the group-average test). We then computed partial correlation-based RSA to test fMRI representations against person-specific models whilst controlling for the corresponding group-average models. Partial correlations were computed on the subset of ROIs that were identified as significant ($p < 0.05$, post FDR adjustment) in the previous group-average model test and the resulting $p$-values were FDR corrected.

**Decoding individual identity from fMRI data**. To further test whether person-specific models could be applied to decode participant identity by identifying the corresponding participants' fMRI data from other peoples', we repurposed an algorithm that was initially introduced to decode word meaning[46]. Two participants were selected at a time, and similarity vectors for the two respective personal models were cross-correlated with fMRI similarity vectors for the two participants. This left four correlation-coefficients associated with both congruent (P1 vs. P1, P2 vs. P2) and incongruent (P2 vs. P1, P1 vs. P2) model vs. fMRI pairings. If fMRI representational structure characterizes individuals, a closer match between an individual's neural data and their own model is expected. To test this, the four correlation coefficients were r-to-z transformed, then the two congruent model-to-fMRI coefficients were summed and the two incongruent coefficients were summed. If the congruent sum was greater than the incongruent sum, decoding was scored as correct (1), otherwise it was scored as incorrect (0). This pairwise match was repeated for every combination of participant pairs (325), and the overall decoding accuracy was quantified as the mean of the 325 scores. Operating at

random, a 50% success rate would have been expected. The statistical significance of decoding accuracies was estimated using permutation testing: similarity vectors for fMRI data only were randomly shuffled (so that participants became linked to other individual's neural data), and the entire decoding analysis was repeated across all 325 pairs. The random shuffling procedure was repeated 10,000 times to generate a NULL distribution of decoding accuracies, and the statistical significance was estimated as the fraction of shuffled accuracies greater than or equal to the unshuffled accuracy.

**Statistics and reproducibility**. fMRI and behavioral data collection were undertaken a single time. In Supplementary Information we demonstrate the robustness of the current analytic approach to changes in parameterization.

**Reporting summary**. Further information on research design is available in the Nature Research Reporting Summary linked to this article.

## Data availability
Source data are provided with this paper. Both preprocessed fMRI and model data to recreate all analyses presented are available at: https://osf.io/2e7f4. https://doi.org/10.17605/OSF.IO/2E7F4. Please contact the authors for (large) raw fMRI datasets. Source data are provided with this paper.

## Code availability
Matlab v2020a code to recreate all analyses and plot main and supplementary figures is available at: https://osf.io/2e7f4.

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

## Acknowledgements

This work was partially supported by the University of Rochester CTSA award number UL1 TR002001 from the National Center for Advancing Translational Sciences of the National Institutes of Health. The content is solely the responsibility of the authors and does not necessarily represent the official views of the National Institutes of Health.

## Author contributions

A.J.A. and F.L. conceived of the experiment. K.M. and A.J.A. collected data. A.J.A. and B. R. analyzed the data. A.J.A. wrote the first draft of the manuscript. D.D.F., K.H., and F.L. contributed to interpreting the results and editing the paper.

## Competing interests

The authors declare no competing interests.
