## [Peer Review File · Nature Communications]

Reviewers' Comments:

Reviewer #1:

Remarks to the Author:

This paper presents a novel method for exploring the neural basis behind memories for everyday events, with an emphasis on exploring the specific memories that different individuals form for similar events (i.e. weddings). I found the paper enjoyable to read, the analysis to be well conducted, and I think that the method and the results that emerged are really interesting and potentially very important. My feeling is overall that this is a well powered investigation into an important and interesting question that is likely to be suitable for publication in Nature Communications. In this review I make a number of suggestions / queries that may facilitate this process.

First, I felt like that the argument about group differences and subject specific approaches could be more nuanced than in this version. In my work on the neural basis of different aspects ongoing experience we usually examine both individual differences and group level patterns, a method which is inevitable in that we know that (a) personality traits influence not just when people mind wander (for example) but (b) it also impact on what they think about. A recent example from our lab (Turnbull et al., 2018, 2019) is a good example of this In this set of studies we examined how different aspects of experience are related to neural processing explicitly from both an individual differences approach (for example, by seeking how variation in functional connectivity at rest predicts different aspects of experience in the lab) as well as examining common patterns of neural activity when these experiences are occurring (a group analysis). We find a high degree of similarity in these approaches – for example, both approaches suggest that the ventral attention network, and in particular the left dorsolateral prefrontal cortex, is linked to the prioritisation of cognition in line with the tasks demands (i.e. it facilitates the occurrence of on task thought when the task is difficult and off task thought when the task is easy). Perhaps more pertinently we find that the default mode network is important for detailed vivid representations of task relevant information in the more demanding working memory task – both in terms of the group level patterns that are detectable, and in individual variation (people who maintain more vivid experiences couple the DMN to visual cortex more). Note that we also found that this pattern holds when we use things like RSA (Sormaz et al., 2018). In this way the starting point of our approach is to seek empirical convergence between both group level and individual differences in the neural basis of different patterns of experience and generally, we find that this view holds well for many of our findings.

It would improve the impact of the paper, therefore, if the authors could help position their work in a context that the reader could see how the links with contemporary research on this topic. To do so they could present a more nuanced perspective on prior attempts to try to combine group level and individual difference perspectives on the neural basis of different features of experience, both in the introduction, as well as the discussion where the authors consider the general implications of their method for understanding how they may generalise to experiences in everyday situations. To be clear, I do not see this as an issue with either the novelty or importance of this paper, but rather that I believe the work would have greater impact if the revision could help readers see how this work fits into the larger literature on this question. In fact it would be important in the future to try to work out how the analysis that the authors can be explored with respect to this question in part because an obvious advantages of the authors approach is that the common space they examine is defined neurally, and so could allow investigating how people may use similar neural systems to produce qualitatively different patterns of cognition, which I think would be really cool.

Second, I wondered if the authors have any evidence with regards to on whether they are decoding different forms of content or different ways to remember. I have thought about how to interpret the

consequences of machine learning approaches in these types of studies. It seems to me that the authors results could either be because people have different memories because of what has happened to them, or that they reconstruct memories in unique ways at the level of a process (they could rely on different codes, or prioritise semantic information for example). I am not sure that whether it the different actual memories or the way the process that they use to remember them can be resolved in the specific instance but it might be worth explicitly acknowledging that both are viable possibilities in the revision. On a related note, I did wonder if the authors could extend their discussion of the importance of the fact that the participants are old matter. I don't think that this is a big issue, however, I did wonder whether it could be pertinent that these individual could rely more on general knowledge than in a younger cohort.

Signed,

Jonathan Smallwood

Reviewer #2:

Remarks to the Author:

Review of NCOMMS-19-39136

Decoding individual identity from brain activity elicited during the recollection of personal experiences

Authors: Anderson, McDermott, Rooks, Heffner, Dodell-Feder, & Lin

SUMMARY.

The reported study investigates whether individual people show unique neural representational geometries when imagining similar events. Participants imagined and described events in response to generic cues (e.g., a shopping scenario) prior to entering the fMRI. Following the simulations, participants also provided 20 experiential ratings of the events. From the verbal descriptions and the described attributes, the authors then estimated the overall (multimodal) predicted similarity of the events. They did this on a person-specific basis as well as on a group-level basis (i.e., group-averaged model). In the MRI scanner, participants repeatedly re-imagined the events. The group-averaged model predicted the structure of neural representations in a set of brain regions. Importantly, the authors show that person-specific models predict the neural similarities in left posterior cingulate cortex (PCC) and precuneus above and beyond the group-average model (excluding the personalized information of the respective person). In a second set of analyses, the authors show that the actual neural similarity structure shows a greater correlation with personalized models as compared to those of other participants in the two regions. Finally, the authors demonstrate that, by itself, only the person-specific verbal model accounts better for the structure of representations in the left PCC. In contrast, the structure in the left precuneus could better be accounted for by participant-specific verbal descriptions of the episodes and the attribute model based on experiential ratings.

OVERALL COMMENTS.

The present manuscript demonstrates that verbal descriptions of events and a large number of experiential ratings can be used to predict the structure of individually unique neural representations. The results are based on state-of-the-art analysis methods, and are presumably of interest to a

specialized readership. In general, we appreciate the sophistication of the analytical approach but suggest that the study lacks a theoretical foundation and is not well grounded in the extant literature on memory and imagination. We do have several additional questions regarding the background, methodology, and presentation of the results. We elaborate on these comments below in detail (in no particular order).

DETAILED COMMENTS.

(1) We are afraid that the manuscript lacks a sufficient conceptual grounding in the extant literature. Imagination and remembering are used interchangeably. While there are a number of similarities between both processes, they also differ in many regards. We would like to encourage a more careful introduction of these terms as well as their use with regard to the present experimental procedure (e.g., Schacter et al., *Nature Reviews Neuroscience*, 2007).

For example, the phrase “recollection of personal experiences” gives the impression that participants remembered specific events that they had actually experienced. However, participants were in fact instructed to imagine events in response to a number of generic cues, and then asked to re-imagine them in the scanner. We think such imagined experience is not the same as actual “personal experience”, even if they are rated as similar to memories.

(2) More specifically, we were surprised that the authors did not refer to a number of recent papers that used similar approaches to investigate research questions related to the content of simulations (e.g., Robin et al., *Journal of Neuroscience*, 2018; Benoit et al., *Nature Communications*, 2019; Silson et al., *eLife*, 2019) or individual versus collective memory (Gagnepain et al., *Nature Human Behavior*, 2019).

(3) Indeed, we find it somewhat challenging to follow the motivation for the reported research. In the introduction, the authors argue that personal experiences are difficult to assess objectively via subjective reports and suggest that this can be overcome by reading out such experiences by examining their neural representations. (However, are personal experiences not inherently subjective?) The manuscript then goes on to use such very subjective personal reports to identify the “objective” neural correlates. How does this contribute to a better assessment of personal experiences in the sense that they are “quantitatively measured and objectively interpreted” (p. 4)?

(4) The authors fractionate the brain in 90 regions-of-interest based on the Automated Anatomic Labelling atlas and include either the 50, 100, or 200 most informative voxels in their final analysis. This approach renders the results dependent on the choice of the atlas as well as the number of voxels. We wonder why the authors did not instead use a searchlight analysis, that makes no assumptions about anatomical boundaries.

(5) The authors focus their main analysis of subject-specific representations on those regions that showed a significant relationship with the group-average model. We have two comments here:

(a) We think the authors may want to present and interpret the set of regions identified by this analysis. Specifically, from the list of the anatomical labels provided on p. 13 and the supplement, we suggest that this is not a random collection of areas. Instead, they seem to resemble the core

recollection and simulation network (e.g., Vilberg & Rugg, 2014; Benoit & Schacter, 2015) or the posterior-medial memory system (e.g., Ritchey, Libby, & Ranganath, 2015).

(b) We wonder whether the authors are concerned that this analysis precludes the identification of possible regions that encode representations that solely correlate with the subject-specific models? If such regions exist, they certainly should be of particular interest for this research endeavor?

(6) RSA is typically performed on parameter estimates derived from a GLM of the brain activity rather than on the raw BOLD signal averaged across time points (e.g., work by Kriegeskorte and colleagues). This procedure has many advantages. For example, it allows to control for noise by including nuisance regressors that code for movements. Including such regressors seems to be particularly important for the current study, that employed an unusually long functional run. In general, we thus wonder whether there is a specific reason the authors chose their more unconventional approach?

(7) More specifically, the authors may want to elaborate on their choice of the specific time window for the analyses. What is the rationale for choosing the 3.5 s delay? Given that participants repeatedly imagined the same scenario that they had already constructed prior to the MRI session, it seems unlikely that accessing that event would take that long. Also, why do the authors then focus on a 5 s time window? We wonder how these rather unconventional methodological decisions might have influenced the results and whether an approach more closely following community standards might have yielded a different outcome.

(8) Related to the previous point: Did the authors use a criterion to exclude participants based on excessive movement? Given that data was collected in a continuous session with a duration of 30 minutes in a sample of "healthy agers", we wonder how much participants moved and how this might have influenced the quality of the data acquired towards the end of the session.

(9) We suggest that the figures could be improved in several ways. We struggled to comprehend these due to the use of different font faces, sizes, weights, as well as the use of red and green (not differentiable for people with color vision impairments). It would also ease the comprehension, if the captions were to provide some more information. As it stands, we struggled to understand the figures without consulting the main text of the manuscript.

(10) We think it is great that this study had been conducted on age groups other than college kids. However, we suggest that the authors should elaborate on their statement "Naturally, we expect that interpersonal differences can be detected in younger adults who often yield cleaner neural signal." What is a cleaner neural signal in this context, and could the authors provide necessary references to back this up?

MINOR COMMENTS.

(1) We were surprised that the authors sacrifice spatial resolution by resampling the data from 2 to 3 mm³, particularly seeing that the analyses are based on spatial patterns of brain activity. Was there a specific reasoning for that?

(2) We note that there are several typos and grammatical glitches throughout the manuscript as well as omissions of single words and incomplete sentences (e.g., "Vice versa for the attributes and attribute ratings." P. 18; "Lastly, that this work presents a step towards future methods that enable features of imagined experiences to be read directly from the brain." P. 20). The manuscript would benefit from careful proof-reading.

Reviewer #3:

None

REVIEWER COMMENTS

Reviewer #1 (Remarks to the Author):

This paper presents a novel method for exploring the neural basis behind memories for everyday events, with an emphasis on exploring the specific memories that different individuals form for similar events (i.e. weddings). I found the paper enjoyable to read, the analysis to be well conducted, and I think that the method and the results that emerged are really interesting and potentially very important. My feeling is overall that this is a well powered investigation into an important and interesting question that is likely to be suitable for publication in Nature Communications. In this review I make a number of suggestions / queries that may facilitate this process.

Thanks very much for reviewing the article Jonathan, and for the positive appraisal and stimulating suggestions!

First, I felt like that the argument about group differences and subject specific approaches could be more nuanced than in this version. In my work on the neural basis of different aspects ongoing experience we usually examine both individual differences and group level patterns, a method which is inevitable in that we know that (a) personality traits influence not just when people mind wander (for example) but (b) it also impact on what they think about. A recent example from our lab (Turnbull et al., 2018, 2019) is a good example of this. In this set of studies we examined how different aspects of experience are related to neural processing explicitly from both an individual differences approach (for example, by seeking how variation in functional connectivity at rest predicts different aspects of experience in the lab) as well as examining common patterns of neural activity when these experiences are occurring (a group analysis). We find a high degree of similarity in these approaches – for example, both approaches suggest that the ventral attention network, and in particular the left dorsolateral prefrontal cortex, is linked to the prioritisation of cognition in line with the tasks demands (i.e. it facilitates the occurrence of on task thought when the task is difficult and off task thought when the task is easy). Perhaps more pertinently we find that the default mode network is important for detailed vivid representations of task relevant information in the more demanding working memory task – both in terms of the group level patterns that are detectable, and in individual variation (people who maintain more vivid experiences couple the DMN to visual cortex more). Note that we also found that this pattern holds when we use things like RSA (Sormaz et al., 2018). In this way the starting point of our approach is to seek empirical convergence between both group level and individual differences in the neural basis of different patterns of experience and generally, we find that this view holds well for many of our findings.

It would improve the impact of the paper, therefore, if the authors could help position their work in a context that the reader could see how the links with contemporary research on this topic. To do so they could present a more nuanced perspective on prior attempts to try to combine group level and individual difference perspectives on the neural basis of different features of experience, both in the introduction, as well as the discussion where the authors consider the general implications of their method for understanding how they may generalise to experiences in everyday situations. To be clear, I do not see this as an issue with either the novelty or importance of this paper, but rather that I believe the work would have greater impact if the revision could help readers see how this work fits into the larger literature on this question. In fact it would be important in the future to try to work out how the analysis that the authors can be explored with respect to this question in part because an obvious advantages of the authors approach is that the common space they examine is defined neurally, and so could allow investigating how people may use similar neural systems to produce qualitatively different patterns of cognition, which I think would be really cool.

Thanks for these suggestions. We've read through the suggested literature with interest.

In response, we have endeavored to construct a new narrative for the revised manuscript that threads together your suggestions with those of reviewer 2. In particular, we have included a dedicated paragraph in the Discussion that seeks to incorporate your above points and the suggested citations, alongside a couple of our related thoughts. In the Introduction we have included a couple of new sentences to set this later Discussion piece up. However, coverage in the Introduction is currently of a more cursory nature. This is largely because we struggled to keep the word count down whilst also incorporating the additional content suggested by reviewer 2, and decided the Discussion was the best place to emphasise your points.

The “dedicated” new paragraph in the Discussion is pasted in below. This is positioned in the context of a previous paragraph that considers putative similarities and differences between episodic and semantic memory, and considers the future benefits of developing a unified approach to modeling episodic and semantic memory. The new paragraph reads as:

It has been suggested⁸⁴ that episodic and semantic memory systems are anchored on the default network⁸⁵. The default network encompasses multiple interwoven systems⁸⁶ that are active in *ongoing thoughts*²⁹ such as remembering, envisioning the future and making social inferences. Studies attempting to characterize ongoing thought have typically applied “multidimensional experience sampling”, where participants repeatedly rate the nature of their current thoughts on multiple dimensions²⁹ (e.g. whether thought is detailed, in images or verbal, related to the past or future...). Not only do within-participant thought samples covary with concurrent default network activity^{30,38}, but individual differences in resting-state network activity profiles predict *trait-level* individual differences in thought profiles sampled outside the scanner at a later occasion⁷⁰, as well as other psychometric traits³⁷. Thus, accepting that ongoing thought includes episodic recollection/simulation, we hypothesize that individual differences in the current fMRI measures/models will to an extent reflect individual differences in thought profiles sampled in the wild, and that both measures will combine to predict individual differences in personality and other stable psychometric traits. We also hypothesize that the current neural measures and verbal/attribute models could provide new ways to predict individual-differences in ongoing thought at a *state-level*, tracking how the “meaning” of internal monologues⁸⁷ and/or the “experience” of daydreaming* unfolds over time, and possibly even may forecast how novel future events will be experienced.

Additionally, we have added an extra sentence in the Introduction, that is embedded within a paragraph that considers fMRI studies of individual differences.

Research has even begun to characterize brain network activity in terms of the nature of individual differences in ongoing thought, reflecting whether current thoughts are detailed, correspond to the past or future, are verbal or in images and so on^{29,30}. However, whilst these studies do demonstrate that individual differences can be discerned using fMRI, it is unclear that the methods utilized to elicit

* See also Refs^{56,88,89} for image-based computational models of visual imagery.

individual-differences (e.g. picture interpretation, resting) and detect them (e.g. exam results, object similarity judgments, ratings on components of thought) would generalize to the current case of imagining oneself in multiple different scenarios when cued by generic prompts such as “a wedding” or “a funeral”.

The other new sentence in the Introduction is embedded in a paragraph considering fMRI studies of group-level differences.

However, whilst all of the previous between-group differences must be built on top of an accumulation of individual-differences (neural features that appear in one group but not the other), **and whilst both group-average and trait-level individual difference studies have revealed the engagement of similar brain networks^{30,37,38}**, group-averages cannot explain detailed differences in event representations between pairs of individuals from within the same group.

Second, I wondered if the authors have any evidence with regards to on whether they are decoding different forms of content or different ways to remember. I have thought about how to interpret the consequences of machine learning approaches in these types of studies. It seems to me that the authors results could either be because people have different memories because of what has happened to them, or that they reconstruct memories in unique ways at the level of a process (they could rely on different codes, or prioritise semantic information for example). I am not sure that whether it the different actual memories or the way the process that they use to remember them can be resolved in the specific instance but it might be worth explicitly acknowledging that both are viable possibilities in the revision.

Good point. We do not have any direct evidence concerning the interplay between episodic reconstruction/suppression processes and memory representations. However, as you say, the current fMRI data are likely to provide a snapshot of both cognitive control and representation.

Whilst it is tempting to suggest that the fMRI activation patterns detected in the current study reflect scenario representations (because this is what the current models sought to capture), the real relationship is unlikely to be so clear cut. In part, this is because the current model estimation (i.e. verbal description and ratings) would have ridden on top of the control processes that were engaged in imagining the scenarios. In addition, between-scenario similarities in control processes may correlate with between-scenario differences in representation.

As concerns updating the manuscript, we have acknowledged that cognitive processes could have played a role in the current results in the new Discussion. This is placed in the context of a new discussion piece on the role of lateral prefrontal cortex (now emphasised following a reanalysis encouraged by reviewer 2), which based on the literature, would seem likely to have been playing role in cognitive control in the current experiment. Specifically, this new Discussion postulates that lateral prefrontal cortex may play a role in activating and deactivating scenario memories as is pasted in below:

Left Dorsolateral prefrontal cortex (DLPFC: mid frontal) also reflected individual differences. DLPFC is typically associated with “cognitive control processes” that regulate thought and actions according to

task goals/personal interests^{73,74,38}. Thus, DLPFC is likely to have helped regulate the activation and suppression of scenario memories on cue, possibly by deactivating memory codes in the hippocampus^{75,76}. It is interesting (though not unexpected²⁰) that DLPFC would encode detailed scenario representations given its putative role in control. One possibility is that these representations were read from other brain regions to help inform task prioritization. Alternatively, representations could be “indirect”, reflecting the differential engagement of control processes invoked in the recall/simulation of different modalities of information. Future work exploring the dynamics of information flow between regions may help to clarify this.

As hinted at in the above new paragraph, we think that it could be illuminating for the future to examine how the relative timing of brain response differs between “cognitive control” brain regions and other regions associated with representation. Whether fMRI is an appropriate medium for such an investigation is unclear, however potentially the relative timing of brain activation could be estimated from the current data via a voxel-wise encoding approach (aka forward modelling) that regresses voxel activation on time-lagged scenario features, and in so doing estimates response delays. For instance, this might reveal different neural response latencies and response durations associated with different model features in different brain regions. For now, we consider this analysis to be beyond the scope of the current work, however we shall conduct some future analyses examining variability in neural response dynamics.

Relatedly, we have also newly made reference to studies demonstrating that connectivity between medial parietal cortex and regions of the temporal lobe reflects the content of memory recall and ongoing thought.

Other recent work has revealed that imagined places, contexts and people can all be discriminated from fMRI activation within MPC¹⁹⁻²³ and that MPC—temporal lobe connectivity reflects the content of ongoing thought⁷⁰ and memory recall²².

On a related note, I did wonder if the authors could extend their discussion of the importance of the fact that the participants are old matter. I don't think that this is a big issue, however, I did wonder whether it could be pertinent that these individual could rely more on general knowledge than in a younger cohort.

Yup. We've now explicitly addressed this potential age-difference in the Discussion.

How the current results extend to younger adults remains untested. We naturally expect interpersonal differences to also be discernable in younger adults' fMRI data (who are less prone to neural atrophy^{96,97}). However, we also anticipate there will be systematic differences associated with socioecological factors, life experience, and neurophysiology⁹⁷. In particular we hypothesize that younger adults' fMRI data will be characterized by the presence of relatively more “episodic” activation features, based on behavioral studies showing that older adults' verbal descriptions of past⁹⁸ and future-hypothetical⁹⁹ scenarios contain more semantic details (not specific in time and place), and fewer episodic details (specific in time and place).

Thanks again for the review.

Reviewer #2 (Remarks to the Author):

SUMMARY.

The reported study investigates whether individual people show unique neural representational geometries when imagining similar events. Participants imagined and described events in response to generic cues (e.g., a shopping scenario) prior to entering the fMRI. Following the simulations, participants also provided 20 experiential ratings of the events. From the verbal descriptions and the described attributes, the authors then estimated the overall (multimodal) predicted similarity of the events. They did this on a person-specific basis as well as on a group-level basis (i.e., group-averaged model). In the MRI scanner, participants repeatedly re-imagined the events. The group-averaged model predicted the structure of neural representations in a set of brain regions. Importantly, the authors show that person-specific models predict the neural similarities in left posterior cingulate cortex (PCC) and precuneus above and beyond the group-average model (excluding the personalized information of the respective person). In a second set of analyses, the authors show that the actual neural similarity structure shows a greater correlation with personalized models as compared to those of other participants in the two regions. Finally, the authors demonstrate that, by itself, only the person-specific verbal model accounts better for the structure of representations in the left PCC. In contrast, the structure in the left precuneus could better be accounted for by participant-specific verbal descriptions of the episodes and the attribute model based on experiential ratings.

OVERALL COMMENTS.

The present manuscript demonstrates that verbal descriptions of events and a large number of experiential ratings can be used to predict the structure of individually unique neural representations. The results are based on state-of-the-art analysis methods, and are presumably of interest to a specialized readership. In general, we appreciate the sophistication of the analytical approach but suggest that the study lacks a theoretical foundation and is not well grounded in the extant literature on memory and imagination. We do have several additional questions regarding the background, methodology, and presentation of the results. We elaborate on these comments below in detail (in no particular order).

Thanks very much for reviewing our article, and for your interest and constructive feedback concerning both the framing and analyses. In light of your comments, we felt that a major overhaul was appropriate on both counts. In brief, for the framing, we have now provided a new narrative for the article that embeds the original content within contemporary episodic memory/simulation literature, whilst also making connections to other related cognitive domains (as guided by reviewer 1). For the analyses, we have revisited our approach and have run some additional analyses (e.g. a searchlight as per your suggestion). The new analyses have further helped to connect the current results up with the episodic/imagination literature. We have additionally modified our figures to make them more legible and hopefully colour-blind friendly. We hope to have addressed your concerns, and believe the article to be much stronger as a result.

DETAILED COMMENTS.

(1) We are afraid that the manuscript lacks a sufficient conceptual grounding in the extant literature. Imagination and remembering are used interchangeably. While there are a number of similarities between both processes, they also differ in many regards. We would like to encourage a more careful introduction of these terms as well as their use with regard to the present experimental procedure (e.g., Schacter et al., Nature Reviews Neuroscience, 2007).

For example, the phrase “recollection of personal experiences” gives the impression that participants remembered specific events that they had actually experienced. However, participants were in fact

instructed to imagine events in response to a number of generic cues, and then asked to re-imagine them in the scanner. We think such imagined experience is not the same as actual “personal experience”, even if they are rated as similar to memories.

Thanks for bringing our ambiguous phrasing to our attention. We have attempted to provide clarification of these issues at various points throughout the manuscript. We have also rephrased the title accordingly to be more specific.

The new title reads:

Decoding individual identity from brain activity elicited in imagining personal experiences of common scenarios

The introduction sets this up as follows:

Memories of past experiences can be activated and relived through imagination and are also thought to be drawn upon heavily to support mental simulation of hypothetical scenarios¹⁻⁵ by piecing together fragments of information acquired across past episodes¹.

This follows on in the next paragraph of the Introduction:

In previous work, functional Magnetic Resonance Imaging (fMRI) studies of brain activity have identified a core distributed network of neuroanatomical regions that are reliably activated during the recollection and/or imagination of different experiences and scenarios⁶⁻⁸. Regions that are activated in episodic recollection and simulation strongly overlap, which suggests that similar neural machinery is engaged in both cases^{1,3-5} (although activation patterns elicited in remembering, and imagining possible past and future events are still distinguishable⁵).

A related part of the new Discussion now reads as:

<snip> Our results are probably best considered to reflect a mixture of recollection and simulation, and even the scenarios that participants reported as being recollections of real events may have reflected a conflation of information across multiple related episodes⁶⁸ (as might be particularly prevalent in our more routine experimental scenarios, such as driving, housework and cooking).

(2) More specifically, we were surprised that the authors did not refer to a number of recent papers that used similar approaches to investigate research questions related to the content of simulations (e.g., Robin et al., Journal of Neuroscience, 2018; Benoit et al., Nature Communications, 2019; Silson et al., eLife, 2019) or individual versus collective memory (Gagnepain et al., Nature Human Behavior, 2019).

Thanks for the guidance here.

As already stated, we have made major changes to the framing of the article that pay particular attention to the literature on episodic recollection and simulation. We have ensured that the new framing spans these

specific citations above, alongside a selection of other newly inserted citations that we hope do justice to the field.

The changes made run throughout the manuscript. Many of these changes will be apparent in our forthcoming responses. However, particularly pertinent to the case at hand, we have written a new paragraph in the Introduction that includes all of the above suggestions.

In previous work, functional Magnetic Resonance Imaging (fMRI) studies of brain activity have identified a core distributed network of neuroanatomical regions that are reliably activated during the recollection and/or imagination of different experiences and scenarios⁶⁻⁸. Regions that are activated in episodic recollection and simulation strongly overlap, which suggests that similar neural machinery is engaged in both cases^{1,3-5} (although activation patterns elicited in remembering, and imaging possible past and future events are still distinguishable⁵). This *core episodic recollection and simulation network* includes regions of medial parietal cortex, inferior parietal cortex, medial prefrontal cortex, and medial and lateral temporal lobe¹⁻¹³. Researchers seeking to decipher what information is represented in brain activity within regions of this network have shown that different types of event can be distinguished from multiple network regions¹⁴⁻¹⁷, as well different components of individual events¹⁸⁻²⁴ (e.g. people, places, objects, space/time of occurrence). However, it has remained unclear whether differences in activation patterns between individuals imagining similar types of event represent anything more than functionally-irrelevant between-subject noise.

(3) Indeed, we find it somewhat challenging to follow the motivation for the reported research. In the introduction, the authors argue that personal experiences are difficult to assess objectively via subjective reports and suggest that this can be overcome by reading out such experiences by examining their neural representations. (However, are personal experiences not inherently subjective?) The manuscript then goes on to use such very subjective personal reports to identify the “objective” neural correlates. How does this contribute to a better assessment of personal experiences in the sense that they are “quantitatively measured and objectively interpreted” (p. 4)?

We agree that the motivation was expressed in a woolly way here – the intended sentiment had been that individual’s verbal reports may neglect key information and are subjectively interpreted by the listener. However, irrespective of this, this statement was indirectly related to the main findings of the manuscript and we have deleted it. We have tightened up the goals and contribution of the revised manuscript to be more direct and consistent in referring to the achievements of the study in: (1) revealing that fMRI signal reflects meaningful interpersonal differences, and (2) in parallel demonstrating that interpersonal differences can be predicted using the current modeling approach.

In particular, the opening paragraph of the introduction now sets up the goals as.

[Goals are] ...to establish that neural activity elicited during imagination captures meaningful differences between individuals’ mental simulations of similar kinds of events, and to devise quantitative methods that can predict the information represented in neural activation.

The closing paragraph of the Introduction now reads:

<snip>we provide evidence that: (1) Neuroimaging can quantitatively measure meaningful individual differences in brain activity elicited as people imagine complex events from their own perspective. (2) The information content of brain activation can be predicted using personalized models derived from verbal descriptions and behavioral ratings of the imagined events. We discuss the potential implications this has for both basic and applied science.

In the Discussion, the opening paragraph now includes the following:

Importantly, this has provided evidence that fMRI can measure brain activity with sufficient signal to quantify meaningful interpersonal differences in the neural representation of complex imagined events. In parallel, we have shown that the representational structure of person-specific fMRI signal can be predicted using the current personalized verbal and attribute models.

In the Discussion, the concluding paragraph now includes the following:

In conclusion, this study has both revealed and predicted meaningful interpersonal differences in fMRI activity elicited as personal experiences were imagined.

(4) The authors fractionate the brain in 90 regions-of-interest based on the Automated Anatomic Labelling atlas and include either the 50, 100, or 200 most informative voxels in their final analysis. This approach renders the results dependent on the choice of the atlas as well as the number of voxels. We wonder why the authors did not instead use a searchlight analysis, that makes no assumptions about anatomical boundaries.

The primary reason for selecting the AAL ROI-based approach was that the first author has found it to be an effective way to detect the presence of fMRI activation patterns of interest using language models (since Anderson et al. NeuroImage 2015), and often more sensitive than searchlight in this respect. This choice echoed our primary interest which was in revealing that person-specific patterns are represented in fMRI data. However, we do agree that searchlight has benefits in narrowing down the locus of patterns of interest. On the flipside, searchlight comes with its own related set of assumptions (e.g. the size and shape of the typically small searchlight volume, whether patterns of interest will adequately fit inside the searchlight without additionally incorporating too much noise, and whether patterns will be in the same place on different peoples' brains). Thus, we consider that ROI-based/searchlight approaches potentially offer complementary strengths regarding sensitivity, and anatomical specificity respectively. In light of this, we ran a follow up searchlight analysis, which in addition to re-identifying medial parietal cortex, helped expose a network of other brain regions that collectively echo the layout of the core recollection/simulation network. We believe that this new searchlight helps greatly to link the current results to the episodic simulation literature, so thanks for nudging us into that. The new section of the results is appended below:

Figure 4. Neuroanatomical distribution of person-specific representational structure: RSA-Searchlight companion to Figure 3. Computation of the three RSAs illustrated here, as well as hypothesis testing and FDR⁶³ correction mirrored the protocol of the previous ROI-based analyses of **Figure 3** conducted within a searchlight framework. Differently ROI selection was by passing a 3-voxel radius cube throughout the brain (rather than segmenting anatomical atlas regions). The heat maps illustrate t-statistics corresponding to one sample t-tests of the corresponding RSA coefficients against zero. The t-statistics illustrated correspond to p-values that survived an FDR threshold placed at $q=0.1$ ($q=0.1$ was used rather than $q=0.05$ to enhance the visibility of clusters for display purposes). The anatomical makeup of clusters arising from an FDR threshold of $q=0.05$ are listed in detail in **Supplementary Tables 3 to 5**.

Detailed neuroanatomical distribution of person-specific representational structure

To follow up our previous ROI-based analysis we performed a “searchlight” analysis⁶⁵ to more precisely estimate the neuroanatomical layout of brain regions reflecting person-specific and group-average models. To this end we replicated the previous RSA and partial RSA within searchlight ROIs. Searchlight ROIs were cubes of radius 3 voxels (side 7) that were iteratively centered on every location in the brain via the implementation in Ref⁶⁶ (as is analogous to shining a searchlight, see **Methods**). This complemented the previous anatomical ROI-based analysis which was well equipped to detect the presence of person-specific information in fMRI data (because informative voxels were selectively analyzed and weaker assumptions were placed on the shape of patterns of interest, beyond that they could fit into the relatively large anatomical ROIs). However, the previous ROI analysis did not precisely locate person-specific representations.

Results of the searchlight analyses are illustrated in **Figure 4**, and the neuroanatomical locations of significant clusters ($p < .05$ FDR⁶³ corrected) are identified in **Supplementary Tables 3 to 5**. Similar to **Figure 3**, fMRI representations in medial parietal cortex and inferior parietal cortex were identified by the group-average models (**Figure 4 right**) and were subsequently found to reflect person-specific information structure in the partial RSA analysis (**Figure 4 left**). Differently, prefrontal regions were not detected and inferior parietal cortex was less well represented (see **Figure 3 left**), which may reflect a lower sensitivity of the searchlight approach (which did not exclude non-informative voxels, and may have been disadvantaged in capturing patterns that did not adequately fit within the searchlight). The searchlight RSA based on person-specific models alone (i.e. when the group-average was not controlled for) revealed a widely distributed network of brain regions that included clusters in dorsal and ventral medial prefrontal cortex, dorsolateral frontal cortex and anterior temporal cortex (**Figure 4 middle**). Importantly, this more neuroanatomically precise estimate of the distribution of regions encoding scenario information echoes the configuration of the core episodic simulation/recollection network⁶⁻⁸ more strongly than **Figure 3**.

Please note also that in the interests of streamlining the article, we have moved the final analysis of our initial article (Old Figure 5) to Supplementary Materials (Supplementary Figure 2) to make way for the new searchlight analysis, which we believe to be more pertinent to the main goals of the article (revealing person-specific representation). To recap, the final analysis of the original article (now Supp Fig 2) demonstrates that the verbal and attribute model both made independent contributions to estimating fMRI activity patterns.

(5) The authors focus their main analysis of subject-specific representations on those regions that showed a significant relationship with the group-average model. We have two comments here:

(a) We think the authors may want to present and interpret the set of regions identified by this analysis. Specifically, from the list of the anatomical labels provided on p. 13 and the supplement, we suggest that this is not a random collection of areas. Instead, they seem to resemble the core recollection and simulation network (e.g., Vilberg & Rugg, 2014; Benoit & Schacter, 2015) or the posterior-medial memory system (e.g., Ritchey, Libby, & Ranganath, 2015).

We agree. We have illustrated the ROIs revealed by the new anatomical ROI analysis (see later) and the new searchlight analysis (above) in the results. The regions exposed by both analyses mirror the layout of the core recollection and simulation network. In the new Discussion we interpret the regions recovered in more detail. This new part of the Discussion is pasted in below.

The current study revealed that fMRI activation patterns in left/right medial parietal cortex, left temporoparietal junction, left dorsolateral prefrontal cortex, ventral and medial prefrontal cortex, and anterior temporal lobe reflected the representational structure of individual participants' personal descriptions/ratings of their imagined scenarios. This distribution of regions closely resembles the layout of the established core episodic recollection and simulation network⁶⁻⁸. Our results are probably best considered to reflect a mixture of recollection and simulation, and even the scenarios that participants reported as being recollections of real events may have reflected a conflation of information

across multiple related episodes⁶⁸ (as might be particularly prevalent in our more routine experimental scenarios, such as driving, housework and cooking).

In particular, we found that medial parietal cortex (MPC: precuneus and posterior cingulate) encoded person-specific representations strongly. MPC is routinely activated in episodic recollection/simulation and also in the perception of scenes and space^{10-13,69}. MPC is believed to play a role in segmenting events from continuous experience and encoding abstract cross modal event representations during recall^{16,17}. Other recent work has revealed that imagined places, contexts and people can all be discriminated from fMRI activation within MPC¹⁹⁻²³ and that MPC—temporal lobe connectivity reflects the content of ongoing thought⁷⁰ and memory recall²². Our study adds to this by revealing that MPC representations of complex imagined events are person-specific and can be predicted using the current verbal/attribute models.

The left temporoparietal junction (LTPJ: posterior mid temporal cortex, inferior parietal cortex, angular gyrus and mid occipital gyrus) also strongly encoded person-specific information. Besides sharing a role in event segmentation and simulation^{16,17}, LTPJ has been linked to the so-called *autonoetic*⁷¹ conscious subjective experience associated with remembering episodes from a first-person perspective^{13,72}, as well as bodily self-consciousness⁷³. Indeed, a novelty of the current approach was the initiative to model imagined scenarios from participants' first-person perspectives. However, whilst this approach proved successful, future work will be necessary to evaluate what role perspective played here. For instance, our participants didn't additionally describe/rate scenarios from a third-person perspective, so we cannot be sure that such a model would not have worked just as well. We hope to elaborate on this in the future.

Left Dorsolateral prefrontal cortex (DLPFC: mid frontal) also reflected individual differences. DLPFC is typically associated with “cognitive control processes” that regulate thought and actions according to task goals/personal interests^{74,75,38}. Thus, DLPFC is likely to have helped regulate the activation and suppression of scenario memories on cue, possibly by deactivating memory codes in the hippocampus^{76,77}. It is interesting (though not unexpected²⁰) that DLPFC would encode detailed scenario representations given its putative role in cognitive control. One possibility is that these representations were read from other brain regions to help inform task prioritization. Alternatively, representations could be “indirect”, reflecting the differential engagement of control processes invoked in the recall/simulation of different modalities of information. Future work exploring the dynamics of information flow between regions may help to clarify this.

The hippocampus, which is a key “hub” of episodic memory¹³ and simulation², was conspicuous by its absence from our results. The hippocampus has been described in terms of providing a “spatial scaffolding” for episodic memory that indexes the neocortical components that code perceptual, emotional, and conceptual content of experience and provide a sense of autonoetic consciousness¹³. In light of this, it is reasonable to suppose that the current attribute/verbal models may better predict the reconstructed neocortical perceptual/emotional/conceptual content than the hippocampal scaffolding. This said, close inspection of our results (**Supplementary Table 1 and 2**) reveals that hippocampal (and parahippocampal) ROIs did significantly reflect personal scenario representations prior to correction for multiple comparisons.

In addition, our searchlight analyses identified clusters in: (1) Ventromedial prefrontal cortex, which has recently been found to encode the identity of known people, places and whether they were liked when fictitious meetings are imagined⁷, and more generally is thought to play a role in encoding self-reference⁷⁸, emotional information⁷⁹, and remote (years-old) episodic memories^{13,15}. (2) Dorsomedial prefrontal cortex, which has been associated with inferring traits of other people⁷⁸, and has recently been shown to encode collective memories reflecting sociocultural group membership²⁴. (3) The anterior temporal lobe, which is thought to be a key semantic memory hub⁸⁰.

(b) We wonder whether the authors are concerned that this analysis precludes the identification of possible regions that encode representations that solely correlate with the subject-specific models? If such regions exist, they certainly should be of particular interest for this research endeavor?

We agree that this is a point of interest that had not been emphasised. The lack of emphasis was a byproduct of the way that we framed and set up the primary analysis – which we still actually like. i.e. Detecting representations of common group-level information structure and then showing that person-specific information structure rests on top of that. This said, we agree that it is beneficial to visually illustrate brain regions that were correlated with the person-specific models (these had initially been concealed in the Supplementary Materials). Additionally, our previous approach to ROI selection was not the only possibility.

With respect to the first point, both the new anatomical ROI analysis (reported in more detail later) and the new searchlight analysis (above) now illustrate the brain regions that reflect the representational structure of the models. The searchlight analysis in particular illustrates comparative results for the person-specific and group-average models.

With respect to the second point, we now consider ROI selection in two different ways, the second of which deploys person-specific representation. Both approaches uncover similar ROIs. Specifically, we apply (1) The initial approach utilized in the original article. (2) We base ROI selection for participant X, on person-specific model vs fMRI correlations for all other participants Y, Z, A, B... It turns out that both approaches identify a similar selection of ROIs (~90% overlap). Due to these negligible differences between the two approaches we prefer to keep with our original approach to simplify interpretation (i.e. approach 2 is more complicated to illustrate and interpret because it identifies a different set of ROIs for each participant).

The above is documented in detail in a new Methods section.

ROI selection based on other participants' data

Because many ROIs were unlikely to represent information associated with the imagined scenarios (i.e. ROIs that are outside the recollection/simulation network), we identified a subset of regions on which to focus our primary individual differences analysis. Our initial interest here was in establishing whether ROIs that represent common group-average scenario structure are further characterized by person-specific idiosyncrasies. This was based on the natural assertion that despite person-specific idiosyncrasies, different people would imagine different scenarios in a broadly similar way (see also Ref¹⁶). We first tested this assertion of cross-participant commonality by performing RSA to compare person-specific multimodal models between each pair of participants. In total this test was performed on 325 pairwise combinations of the 26 individuals. This yielded a mean \pm SD Spearman's correlation coefficient of 0.33 ± 0.11 , which when r-to-z-transformed was significantly greater than zero ($t=52.8$, $p<1e-128$). This supported our assertion that there were indeed strong commonalities in scenario representation between individuals.

Then, we isolated ROIs that reflected group-average representations across participants to later test for person-specific information. We ran RSA between each ROI and group-average models (that excluded the test participant) for each test participant. This produced an RSA correlation coefficient for each ROI and each participant. All correlation coefficients were r-to-z transformed. Then, for each separate ROI, the set of coefficients corresponding to the 26 participants were compared to zero using one sample t-tests (1-tailed, anticipating positive correlation). This yielded a vector of 90 p-values. The 90 p-values were corrected for multiple comparisons according to False Discovery Rate⁶³ (FDR). FDR corrected p-values corresponding to eight ROIs were under the conventional $p=0.05$ threshold. These were: Precuneus_L, Precuneus_R, Cingulum_Post_L, Occipital_Mid_L, Parietal_Inf_L, Temporal_Mid_L, Frontal_Mid_L and Angular_L. We considered these eight ROIs provisionally as candidates to take forward to the next person-specific test.

However, before continuing forward, we tested whether this selection of ROIs was reasonable from a different perspective. The previous ROI selection approach risks overlooking ROIs that might solely correlate with person-specific models (and not the group-average models) and besides, also carries a risk of emphasizing effects associated with the group-average models (if group-average models happen to fit fMRI noise that is not subsequently captured by the personal models). Thus, we further tested ROI selection based on comparisons between other participants' fMRI data and their personal models. Specifically, to select ROIs for each participant (e.g. P1), we computed RSA between each other participant's (e.g. P2 to P26) fMRI data and their personal model, which was repeated for each ROI. RSA correlation coefficients were r to z transformed. For each ROI, the set of 25 (n-1) coefficients

were compared to zero using one sample t-tests (one-tailed, anticipating positive correlation). This yielded a vector of 90 p-values. The 90 p-values were FDR corrected and ROIs with adjusted p-values < 0.05 were selected. This resulted in 7/8 of the previously selected ROIs being identified in every one of the 26 participants. The left-out ROI was Angular_L which was selected in 20/26 participants. In addition, Frontal_Med_Orb_L, Frontal_Sup_L were identified for every single participant, alongside a mixture of other ROIs that differed between participants. To simplify the interpretation and visualization of results we performed our forthcoming analyses on the original fixed set of eight ROIs (rather than a different subset of ROIs for each individual). A comprehensive listing of test statistics for all 90 ROIs is in **Supplementary Table 1**. Frontal_Med_Orb_L, Frontal_Sup_L are later identified in our searchlight analysis **Figure 4**.

(6) RSA is typically performed on parameter estimates derived from a GLM of the brain activity rather than on the raw BOLD signal averaged across time points (e.g., work by Kriegeskorte and colleagues). This procedure has many advantages. For example, it allows to control for noise by including nuisance regressors that code for movements. Including such regressors seems to be particularly important for the current study, that employed an unusually long functional run. In general, we thus wonder whether there is a specific reason the authors chose their more unconventional approach?

The current approach was in one part borne out of personal habit – we have found averaging to be a successful approach in related work studying the neural correlates of active thought and imagination (since Anderson et al. *NeuroImage* 2015, Anderson et al. *JoCN* 2014). We initially borrowed the approach from Mitchell et al. *Science* 2008, and observed that averaging was used in a recent related work that discriminated fMRI activation elicited by suicidal ideators and healthy controls contemplating emotional concepts (Just et al. *Nat Hum Behav* 2018)

In another part, the averaging approach was borne out of our having low confidence that participants' brains would all exhibit canonical response profiles as they undergo the current task (which relies on them to consciously coordinate their recollections/imagination, as opposed to say testing for automatic perceptual responses). Such commonalities would be assumed by canonical HRF-based GLM approach. Specifically, the current participants were responsible for consciously bringing their specific scenario to mind, reliving the scenario vividly and then subsequently suppressing it from imagination. In undertaking related experiments even we as dedicated participants have had difficulty in switching our imagination on and off in time with an experimental prompt. Thus, we were not confident that our current participants would fare much better in this respect in the current experiment.

However, given your concerns we were inclined to do some additional investigation into (1) our assertion that the assumptions of a canonical HRF approach would not be met (i.e. that neural responses will not be at the same latency and duration for all participants) and (2) relatedly whether the parameters that we had selected for the averaging window were well chosen. This is continued in our response to your next query.

(7) More specifically, the authors may want to elaborate on their choice of the specific time window for the analyses. What is the rationale for choosing the 3.5 s delay? Given that participants repeatedly imagined the same scenario that they had already constructed prior to the MRI session, it seems unlikely that accessing that event would take that long. Also, why do the authors then focus on a 5 s time window? We wonder how these rather unconventional methodological decisions might have influenced the results and whether an approach more closely following community standards might have yielded a different outcome.

These parameters were initially chosen at the time of running a preliminary analysis on a pilot participant (and they seemed reasonable at the time!). As it turned out this pilot set up immediately scaled up to the entire study group to yield the results of the original manuscript. We were very satisfied with these results as a baseline and didn't try to further manipulate the parameters.

Irrespective, to address your points (6) and (7) we have now run some additional analyses to estimate participants' hemodynamic responses characteristics from the current data, and more pertinently whether responses differed between participants. Based on these analyses it would seem that: (1) The dynamics of different participants' hemodynamic responses varied substantially, both in terms of the latency and duration of response. (2) Our initial parameter selection for the averaging window was not particularly astute. This is because some participants' hemodynamic responses peaked around 5secs (before our original analysis window), and other participants had high activation in the volume after our analysis window.

In light of (1) we prefer to maintain our activation averaging approach. This is because we remain cautious about accepting the assumptions of a canonical HRF-based approach for the current experiment (see previous response and later) because different participants' hemodynamic responses appear to be out of phase with canonical HRFs to varying degrees. Importantly, the current averaging approach has more relaxed assumptions in this respect because it does not predict the timing of the peak hemodynamic response and its duration.

In light of (2), we have changed our analysis window to four volumes (10sec) beginning 5secs after stimulus onset. Thus, this now includes one volume before the original analysis window and one volume after. To recap, participants were instructed to imagine their scenario whilst the written stimulus prompt was displayed on screen (and thus for a period of 7.5 seconds equating to three volumes). Thus, our new analysis window can be considered to reflect neural activity in the period ~ 0 to 10seconds after stimulus display. This time window assumes that participants did not clear their minds instantly at 7.5seconds (when the stimulus disappeared). This seems reasonable based on our own experience of trying to clear our mind in similar experiments and also our HRF analysis in response to point (1).

The result of this change in analysis window was an improved sensitivity in our analyses. Not only did the reanalysis emphasise the engagement of Precuneus and Posterior Cingulate (replicating our original results), but it also newly revealed representations in other regions of the core episodic/recollection network.

For example, the new Figure 3 illustrating the results of the ROI analysis is pasted in below:

Figure 3. fMRI activation patterns elicited in imagining personal experiences reflect person-specific information. The critical plot is at the top right. This plot demonstrates that participant-specific scenario models predict the representational similarities of corresponding fMRI data over and above group-average models (that exclude the respective participant). Open black circles illustrate RSA coefficients for the 26 participants. Bar heights correspond to mean values across participants. t-tests tested whether RSA coefficients were greater than zero. Cohen's d was computed by dividing the t-statistic by $26^{1/2}$. i^* identifies the number of individual-level RSA permutation p-values < 0.05 (maximum 26, see **Methods** for details). The mid-right and bottom right plots illustrate RSA comparisons of fMRI data with person-specific models and group-average models respectively. The eight ROIs presented were selected according to the procedure described in **Methods**. FDR⁶³ correction for person-specific and group-average models (mid and bottom plot) was across 90 ROIs. FDR correction in the partial RSA (top) was across eight ROIs. A detailed listing of results for all ROIs is provided in **Supplementary Tables 1 and 2**. Brain illustrations were made using Ref⁶⁴.

A byproduct of analysis using the new analysis window was a small change in the outcome of the original article's final analysis (Old Figure 5). This analysis had initially demonstrated that the verbal model alone was sufficient to explain neural representations in Left Posterior Cingulate and both models contributed to the Precuneus. Now both models contribute to explaining representations across both the Precuneus and Posterior Cingulate, as well as the other six ROIs plotted in **Figure 3**. This analysis is now presented in **Supplementary Figure 2**, to make way for the new Searchlight analysis (as we have already outlined in an earlier response).

We have included the rationale for our averaging approach in the Methods below:

Besides having observed averaging activation to be an effective strategy in other researchers' studies of active thought^{35,46} and in our own work on mental simulation/active thought^{56,88,100}, we chose the averaging approach in favor of a canonical HRF-based approach because we were not confident that some of the assumptions made by the canonical approach would hold for the current task (which relies on participants to consciously coordinate their recall/imagination, as opposed to say testing for automatic perceptual responses). In particular, the canonical HRF approach would both assume and predict precisely the same neural response profile (including peak response latency and response duration) for each participant and each scenario. This would be modeled by convolving a time series marking the stimulus display periods with a fixed canonical hemodynamic response function. We were initially cautious about this assumption based on our own experience as dedicated participants undertaking different mental simulation studies of a similar ilk, where we have found it difficult to rigorously synchronize our conscious imagination to stimulus onsets and disappearances. To follow up on these hunches we performed a preliminary investigation into the degree of interpersonal variation in hemodynamic responses for the current data.

Results of this investigation are included in **Supplementary Figures 19 to 21**. Qualitative inspection suggests that there was indeed substantial interpersonal variability in both the latency and duration of participants' hemodynamic response profiles. For example, some participants appear to have responded rapidly in imagining scenarios and also appear to have cleared their minds relatively sharply. Other participants appear to have responded more slowly and to have kept mental simulations going on for longer. Thus, modeling each participants' activation using the same HRF would have captured some participants' neural response profiles well and others poorly. As the goal of the current study was to capture interpersonal variation, we opted for the averaging approach in attempt to avoid biasing our results to reflect only the subset of participants who exhibit canonical responses. We do plan to pursue the nature of these interpersonal (and potentially inter-scenario) differences in neural response dynamics in future work. However, for the time being the current averaging approach provides a practically effective solution to the current problem that lacks the assumption of common peak response latencies and durations across participants.

A snapshot of the HRFs estimated for individual participants and a description of the analyses is pasted in below. Complete Results for all participants are in Supplementary Figures 20 and 21.

Supplementary Figure 19. Different participants were characterized by different hemodynamic response functions (HRF). We conjectured that for the current episodic simulation task HRFs would differ between people. If true this would challenge the validity of modeling the current fMRI data using the same canonical HRF for each person. To test this, we separately estimated HRFs for each individual using multiple regression to predict voxel activation based on a time-lagged stimulus representation*. The resultant beta-weights at different time lags estimate the HRF unfolding over time. We explored two ways of modeling the visual stimuli: (1) as an onset “spike” (top left), or as a “boxcar” reflecting when the stimulus was on display (bottom left). HRFs were separately estimated for each of the 20 scenarios, within each run. First, a separate stimulus timeseries was created for each scenario, within each run, at the same sample rate as fMRI (2.5sec). Ones were entered to mark stimulus display (spike/boxcar), the rest of the vector was zeros. To account for hemodynamic delays the vector was copied 6 times (reflecting the inter-stimulus interval), and each copy was temporally offset by one TR greater than the previous. Thus, if vector 1 had a one in position three, then vector 2 would have a one in position four. The vectors were concatenated into a 6column matrix. 6 head motion parameters and linear trend were concatenated with this matrix. Voxel activation time series were also represented as column vectors. Both fMRI and stimulus matrices were normalized so that each column had mean 0 and SD 1. To estimate person-specific HRFs, each voxel’s activity was separately regressed on the stimulus matrix. This was repeated for each stimulus (scenario per run). To counteract overfitting, ridge regression was used (penalty=1). Beta-weights estimated the magnitude of each voxel’s hemodynamic response across the 6 volumes post stimulus onset. Beta weight profiles were averaged across all scenarios and then across all voxels in Left Precuneus (which yielded strong results in our main analyses e.g. **Figure 3**). Participants’ estimated HRFs are plotted in black. Overlaid in red is a canonical HRF computed by convolving a spike/boxcar with `spm_hrf(2.5)`. Qualitative inspection suggests that different participants were characterized by different HRF latencies and durations. Critically, the canonical “spike” HRF weakly reflected the HRF estimated for P4 to P6. The canonical “boxcar” HRF was a poor match for P1 to P4. This provides evidence that using one canonical HRF to model all participants would result in a weak fit to many participants data. See **Supp. Figures 20/21** for complete results.

* For an illustration of a similar approach see: Broderick MP, Anderson AJ, Di Liberto GM, Crosse MJ, Lalor EC. 2018. Electrophysiological correlates of semantic dissimilarity reflect the comprehension of natural, narrative speech. *Current Biology*. 28(5):803-9.

(8) Related to the previous point: Did the authors use a criterion to exclude participants based on excessive movement? Given that data was collected in a continuous session with a duration of 30 minutes in a sample of “healthy agers”, we wonder how much participants moved and how this might have influenced the quality of the data acquired towards the end of the session.

No, we didn't exclude participants. In fact, the vast majority of participants appear to have remained relatively stationary during the scan, though as you surmise head displacement was on average greatest towards the end of scanning. We have also included a figure illustrating head motion parameters in the Supplementary Materials (pasted in below). We did not exclude participants because there were relatively a few cases of high motion, and it was unclear whether the periods of head motion observed would compromise the corresponding participant's entire data set (though admittedly #23 is questionable and did not yield a significant model vs fMRI RSA result at individual-level). We have however now regressed out head motion parameters from the fMRI data prior to analysis (see below).

Six head motion parameters (translation on x,y,z axes, and yaw, pitch and roll, see **Supplementary Figure 18**) and linear trend were voxel-wise regressed out from the fMRI data within each of the 5 runs. Specifically, prior to regression, each vector of voxel activation, and each vector of the previous nuisance regressors was z-scored within each run by subtracting the vector-wise mean and dividing by the vector-wise standard deviation. A separate multiple regression was computed to predict voxel activation from the nuisance parameters. Finally, the residuals arising from the regression for each voxel were taken forward to use in computing the scenario representations that would form the basis of our subsequent analyses. Individual scenarios were represented by voxel-wise averaging across the four fMRI volumes (of residuals) spanning the period 5 to 15 seconds after stimulus onset (at 15secs the next stimulus prompt was displayed). The point of onset at 5 seconds was selected because the hemodynamic response associated with imagination was likely to peak at the very earliest at 5 seconds post stimulus presentation (the visual response to the written stimulus would be expected to peak at around 5secs, with recall/imagination following later). The interval of 4 volumes were measured to allow for between participant and between scenario differences in hemodynamic response profiles. Such hemodynamic differences could arise from a mixture of differences in neural response latencies associated with consciously retrieving scenarios from memory and bringing them to mind, reliving the multisensory experience of the simulated scenario (which could play out for different durations for different scenarios/people), and subsequently suppressing the imagined scenario.

Supplementary Figure 18. Estimated head translation and rotation across the duration of the experiment.

In response to the query concerning the contribution of data in the final run, we also quickly reran an RSA analysis from scratch to see what would happen if we deleted fMRI data associated with the last run (so the analysis was run on runs one to four rather than one to five). The outcome of this analysis is illustrated **right**.

The Figure (right) is in the same format (and uses the same ROI colour codes) as Figure 3. Sorry about the small font size on the x-axis - this Figure is not included in the main manuscript, but hopefully the results are still clear.

In summary, on face value excluding run 5 does not appear to have a dramatic effect on the outcome. So, any left-over effects of motion-related noise towards the end of the scan are at least not disrupting results substantially.

(9) We suggest that the figures could be improved in several ways. We struggled to comprehend these due to the use of different font faces, sizes, weights, as well as the use of red and green (not differentiable for people with color vision impairments).

It would also ease the comprehension, if the captions were to provide some more information. As it stands, we struggled to understand the figures without consulting the main text of the manuscript.

Thanks for letting us know. We have updated all of our figures (see the previous responses) and hope that they are more legible.

The new ROI colour-scheme is now in theory friendly for protanopia, deuteranopia, or tritanopia. Relatedly, we hope that Figures 1 and 2 (that illustrate the experimental protocol and our analytic approach) are also clear. Red and green are used in these figures admittedly. However, knowing these colours is not necessary for interpretation (because we have iconized the different models). The figures were clear to us at least when printed in black and white.

Otherwise, we have been consistent in using Arial font across our Figures. There is one exception. This is for the “hand-written” script in Figure 1, where we intentionally chose different fonts for different participants to convey the impression of individual differences.

Finally, we have extended the captions of all figures to be more descriptive.

(10) We think it is great that this study had been conducted on age groups other than college kids. However, we suggest that the authors should elaborate on their statement “Naturally, we expect that interpersonal differences can be detected in younger adults who often yield cleaner neural signal.” What is a cleaner neural signal in this context, and could the authors provide necessary references to back this up?

Indeed, we could have phrased that better. We have rewritten the paragraph in question, whilst also taking care to address a related comment made by reviewer 1.

How the current results extend to younger adults remains untested. We naturally expect interpersonal differences to also be discernable in younger adults’ fMRI data (who are less prone to neural atrophy^{96,97}). However, we also anticipate there will be systematic differences associated with socioecological factors, life experience, and neurophysiology⁹⁷. In particular we hypothesize that younger adults’ fMRI data will be characterized by the presence of relatively more “episodic” activation features, based on behavioral studies showing that older adults’ verbal descriptions of past⁹⁸ and future-hypothetical⁹⁹ scenarios contain more semantic details (not specific in time and place), and fewer episodic details (specific in time and place).

MINOR COMMENTS.

(1) We were surprised that the authors sacrifice spatial resolution by resampling the data from 2 to 3 mm³, particularly seeing that the analyses are based on spatial patterns of brain activity. Was there a specific reasoning for that?

Practicalities dictated this step, but we believe it made virtually no difference to the results. In pilot analyses on a couple of participants we found there to be a tiny boost in correlation coefficient associated with performing analyses on 2mm³ rather than 3mm³ voxels. We downsampled to 3mm³ so that the data would fit onto an already cluttered hard drive to facilitate analyses.

(2) We note that there are several typos and grammatical glitches throughout the manuscript as well as omissions of single words and incomplete sentences (e.g., “Vice versa for the attributes and attribute ratings.” P. 18; “Lastly, that this work presents a step towards future methods that enable features of imagined experiences to be read directly from the brain.” P. 20). The manuscript would benefit from careful proof-reading.

Thanks for spotting them, we’ve tried to iron those out.

Reviewers' Comments:

Reviewer #1:

Remarks to the Author:

This revised paper deals with all of the issues I raised in the initial round of reviews. I think that the changes to the way the results are explained, as well as the additional analyses in this revision strengthen the paper. Reading it through it is an excellent piece of work and I would like to congratulate the authors on an excellent piece of work.

Signed

Jonathan Smallwood

Reviewer #2:

Remarks to the Author:

Review of NCOMMS-19-39136A

We first want to apologize for the delayed response caused by the pandemic.

We think that the authors have been very responsive to our comments and suggestions. Though we continue to think that the strength of the project is primarily its methodological approach, it is now much better embedded in the extant literature. Overall, the manuscript has become much clearer, particularly also the motivation and unique contribution of the study. We thus have a few remaining recommendations:

(1) At times, the manuscript still uses the terms "imagining" and "remembering" rather interchangeably. For example, in the introduction (p. 3, L. 6), when referring to the reliving of past events, the term "remembering" seems more suitable than imagining. The same holds for the abstract: "This leads to personal memories that presumably provide neural signatures of individual identity when events are imaged."

(2) We appreciate the reasoning regarding the possible inadequacy of using the canonical HRF and think that the authors' approach is appropriate. However, we would suggest that including the temporal derivative of the HRF would possibly have constituted a more principled approach that could even have accounted for individual differences.

(3) Finally, we still think that there are some opportunities for optimizing the figures. In particular, Figure 3 and the respective supplementary figures that use the same layout are very busy and hard to understand without consulting the method section in detail. (Please also note a probably erroneous cross reference to this figure on p. 12).

Reviewer #1 (Remarks to the Author):

This revised paper deals with all of the issues I raised in the initial round of reviews. I think that the changes to the way the results are explained, as well as the additional analyses in this revision strengthen the paper. Reading it through it is an excellent piece of work and I would like to congratulate the authors on an excellent piece of work.

Signed

Jonathan Smallwood

Thanks for reviewing the article, and we're glad that you found it interesting!

We first want to apologize for the delayed response caused by the pandemic.

Thanks for finding the time to conduct the review in these troubled times.

We think that the authors have been very responsive to our comments and suggestions. Though we continue to think that the strength of the project is primarily its methodological approach, it is now much better embedded in the extant literature. Overall, the manuscript has become much clearer, particularly also the motivation and unique contribution of the study. We thus have a few remaining recommendations:

Thanks for the positive remarks and recommendations.

(1) At times, the manuscript still uses the terms “imagining” and “remembering” rather interchangeably. For example, in the introduction (p. 3, L. 6), when referring to the reliving of past events, the term “remembering” seems more suitable than imagining. The same holds for the abstract: “This leads to personal memories that presumably provide neural signatures of individual identity when events are imaged.”

Thanks, we've made a few alterations in attempt to iron this out. In doing so, we've tried to steer close to using the term imagine in ambiguous cases (to cohere with the experimental instruction “to imagine...”) whilst emphasizing that imagination draws on past memories, and bearing in mind that the current experiment also involves a strong component of recollection.

Most of these changes were made in the abstract and first paragraph of the Introduction. There are a couple of other changes of a similar ilk made elsewhere in the manuscript (highlighted in blue font).

“Abstract

Everyone experiences common events differently. This leads to personal memories that presumably provide neural signatures of individual identity when events are reimagined.”

“Introduction

Almost everyone can imagine themselves at a wedding, however each person does so differently because they have been to different weddings and hence draw upon memories that no one else has. Our personal history of episodic memories contributes to defining us as individuals and in extreme cases – where memories are of traumatic events – can profoundly affect our psychological health and quality of life. A principal goal of cognitive science is to understand how such memories are represented and manipulated in the human brain. Memories of past experiences can be activated and relived through recollection and are thought to be pieced together to support the mental simulation of hypothetical scenarios¹⁻⁵. Functional brain imaging now enables the systematic study of brain activation elicited during recall and imagination. A long-term vision for the future might therefore be to devise technology that provides a comprehensive neural readout of the information that one voluntarily activates in mental imagery. More humble prerequisites for this are to establish that neural activity elicited during imagery captures meaningful differences between different individuals’ mental simulations of similar kinds of events, and to devise quantitative methods that can predict the information represented in neural activation.”

(2) We appreciate the reasoning regarding the possible inadequacy of using the canonical HRF and think that the authors’ approach is appropriate. However, we would suggest that including the temporal derivative of the HRF would possibly have constituted a more principled approach that could even have accounted for individual differences.

Thanks for the recommendation here. Because we believe the current results to already be strong (with respectable effect sizes) and because finding out whether including the temporal derivative would improve them would entail an extensive reanalysis effort (i.e. repeating and replotting all analyses) we considered it expeditious to leave this over as an avenue for future research.

We have explicitly acknowledged that the current approach, whilst being practically effective, may be suboptimal. We have additionally explicitly stated that including the temporal derivative of the HRF may provide future benefits. This change is in the MRI Preprocessing section of the Methods, pasted in below.

“However, for the time being the current averaging approach provides a practically effective solution to the current problem that lacks the assumption of common peak response latencies and durations across participants. For the future, we note that there may be benefits to including the temporal derivative of the HRF in computation, as was suggested by a reviewer.”

(3) Finally, we still think that there are some opportunities for optimizing the figures. In particular, Figure 3 and the respective supplementary figures that use the same layout are very busy and hard to understand without consulting the method section in detail. (Please also note a probably erroneous cross reference to this figure on p. 12).

Thanks for pointing that out and spotting the cross-reference. We’ve split apart the old Figure 3 into two figures (Figures 3 and 4). Figure 3 now displays the brain map and the critical partial-RSA result indicating that the person-specific models predicted fMRI representational structure over and above the group-average models. Figure 4 now displays the other two plots that were initially also included in Figure 3 (the RSAs using the personal model in isolation and the group-model in isolation). We have modified the Results text accordingly to accommodate this new presentation.

We have kept the Supplementary Figures in the original format to conserve space. Whilst we agree that each individual plot would have been easier to view if the figures had also been split up, we did not think this was necessarily advantageous here. We consider the primary purpose of the Supplementary Figures to be to provide a quick confirmation that we get similar results when using various different analytic parameterizations (as opposed to presenting new information). So, the key results are presented in the now hopefully easy to view Figures 3 and 4, and the supplementary figures hopefully facilitate a quick verification that these findings are robust to parameter change.